# Kdm4a is an activity downregulated barrier to generate engrams for memory separation

Xiuxian Guo[1], Pengfei Hong[1], Songhai Xiong[1], Yuze Yan[1], Hong Xie [2] ✉ & Ji-Song Guan [1,3] ✉

Memory engrams are a subset of learning activated neurons critical for memory recall, consolidation, extinction and separation. While the transcriptional profile of engrams after learning suggests profound neural changes underlying plasticity and memory formation, little is known about how memory engrams are selected and allocated. As epigenetic factors suppress memory formation, we developed a CRISPR screening in the hippocampus to search for factors controlling engram formation. We identified histone lysine-specific demethylase 4a (Kdm4a) as a negative regulator for engram formation. Kdm4a is downregulated after neural activation and controls the volume of mossy fiber boutons. Mechanistically, Kdm4a anchors to the exonic region of *Trpm7* gene loci, causing the stalling of nascent RNAs and allowing burst transcription of *Trpm7* upon the dismissal of Kdm4a. Furthermore, the YTH domain containing protein 2 (Ythdc2) recruits Kdm4a to the *Trpm7* gene and stabilizes nascent RNAs. Reducing the expression of Kdm4a in the hippocampus via genetic manipulation or artificial neural activation facilitated the ability of pattern separation in rodents. Our work indicates that Kdm4a is a negative regulator of engram formation and suggests a priming state to generate a separate memory.

Memory engrams are widely detected in many brain regions, where distinct cell ensembles are actively involved in memory encoding, retrieval and separation[1]. It remains an open question: what determines and allocates cell ensembles to encode contextual memory for a new event? Molecular signatures of engram populations after the enrollment have been extensively acquired by genetically trapping learning-activated neurons[2–5]. While transcriptional signature and epigenetic landscape of single granule cell predicts reactivated neurons during memory recall[4,6], the factors determining the engram population before learning remain unresolved. The memory allocation hypothesis suggests that neurons with temporarily increased excitability biases the representation of a subsequent memory within the pre-determined population[7–10] due to a time-dependent CREB activation mechanism[11,12]. Besides event-driven factors, it will be interesting to understand if engram population is also pre-determined by genetic

or epigenetic factors. Previous studies have identified epigenetic factors such as Hdac2[13], Tet1[14], Dnmt1[15] and Suv39h1[16] as critical regulators for the formation, consolidation and extinction of memories. We reasoned that epigenetic factors might play a pivotal role for the memory engram enrollment and could modulate cells into a priming state, readily for memory allocation.

Epigenetic factors emerge as one of the key players in memory regulation. On the one hand, neuronal activity dramatically trigger genome-wide DNA methylation[17], histone modifications[18] and chromatin accessibility[19] changes. On the other hand, alternation of epigenetic factors modulates synaptic plasticity and neural circuit stability by influencing the expression of synaptic plasticity-related genes, synaptic protein and triggering alternative splicing of synaptic proteins[20–22]. Consequently, mutations of epigenetic factors have been identified in memory-related brain disorders, including epigenetic

[1]School of Life Science and Technology, ShanghaiTech University, Shanghai, China. [2]Institute of Photonic Chips, School of Artificial Intelligence Science and Technology, University of Shanghai for Science and Technology, Shanghai, China. [3]State Key Laboratory of Advanced Medical Materials and Devices, ShanghaiTech University, Shanghai, China. ✉e-mail: hongxie@usst.edu.cn; guanjs@shanghaitech.edu.cn

abnormalities in Alzheimer's Disease (AD)[23] and CHD8[24], ASH1L[25], and MECP2[26] in Autism Spectrum Disorder (ASD). It is still unknown whether crucial epigenetic regulators involve in memory allocation and subsequently influence brain diseases.

The *EGR1-EGFP* transgenic mice have been used to report accumulative neuronal activity[27] and track engram activity in vivo[28]. Here, we have built an in situ screening system that utilizes the robust florescence signal as the indicator of engram cells in the hippocampal dentate gyrus and a CRISPR-Cas9-based gene-knockout library to screen for epigenetic factors involved in the generation of memory engrams. A novel factor, Kdm4a, was identified as a key player in engram generation and memory allocation. Kdm4a regulates synapse via controlling the expression of synaptic proteins, such as Trpm7. Interestingly, Kdm4a induces $N^6$-methyladenosine (m⁶A)-mediated stalling of nascent RNAs, which are further stabilized by Ythdc2. This process allows a burst expression of *Trpm7*. Downregulation of Kdm4a expression in neurons converts them into a priming state biased for engram formation during memory allocation. Notably, deleting Kdm4a in neural ensembles promotes the generation of dissociated engrams within 6 h after prior learning and increases pattern discrimination abilities.

## Results

### Kdm4a is a negative regulator to allocate engram in the hippocampus, as identified by an Epifactor-targeted CRISPR screening

To determine which epigenetic regulators are involved in memory allocation, we designed a CRISPR-sgRNA-based in situ screening system in the mouse hippocampus during contextual fear learning (Fig. 1a). Firstly, a sgRNA library targeting 117 reported histone methylation-related genes was synthesized (Supplementary Data 1). In this library, ten single-guide RNAs (sgRNAs) per gene were designed, targeting each of these hit genes. Besides, 15 non-targeting sgRNAs were included in the library as negative control samples, resulting in 1185 sgRNAs in total. The sgRNA library was cloned into the lentiviral CRISPRv2-mCherry backbone[29] (Supplementary Fig. 1a, b) and packaged into high titer lentivirus. The dosage was adjusted to ensure sufficient expression and coverage of sgRNA from the library in vivo based on previous CRISPR-based screening[30,31] (Supplementary Fig. 1c-f).

Next, *Egr1-EGFP* reporter mice, which express green fluorescent protein in activated neurons[16,28] (Supplementary Fig. 1g-i), were used to identify the engaged engram ensembles in the dentate gyrus (DG). About 3 weeks after delivering the Epifactor lentiviral CRISPR-ko library into the DG, *Egr1-EGFP* mice were subjected to contextual fear conditioning to activate engrams in the hippocampus. DG neurons were dissected at 1.5 h after conditioning. After collecting the infected mCherry⁺ neurons from 22 mice (about 2000 neurons per mice), we evaluated the relative sgRNA enrichment in the engram population (mCherry⁺EGFP⁺) and the non-engram population (mCherry⁺ EGFP⁻) (Fig. 1b and Supplementary Fig. 1j-n and Supplementary Fig. 2) to determine which epifactor knockout was dispensable for the cells to become part of the engram. Lentivirus-infected cells were mainly found in the DG granule cells (GCs) (> 90%), with less than 10% in CA3 or Hilus (Supplementary Fig. 3a-c). A few inhibitory neurons (< 1%) and glial cells (< 3%) were also labeled (Supplementary Fig. 3d-l), but they might not have a strong impact on this screening due to the limited number of these cells. As a control experiment, we did not detect significant enrichment of a particular cell population, as the expression of cell-specific markers between mCherry⁺EGFP⁺ and mCherry⁺EGFP⁻ populations (Supplementary Fig. 4). Among the 117 epifactors, several genes, including *Kdm4a, Kdm4d, Kdm6a, Zmynd11* and *Cdc73*, emerged as candidates in our initial in situ screening (Fig. 1c). They were enriched (fold change > 1, $p < 0.05$) in the engram population compared to the non-engram population (Supplementary Data 2). Only *Mecp2* and *Kdm4a* showed significant difference considering all targetting

sgRNAs (Fig. 1d). As a control, the non-targeting sgRNAs did not show any preference between the engram and non-engram population (Fig. 1c, d and Supplementary Fig. 1j-n). Interestingly, *Mecp2*, which modulates neuronal activity[32], also showed enrichment in the engram population (Fig. 1d).

Among all the candidates, we chose *Kdm4a* for further studies as it showed the best robust and statistically significant phenotypes in many validation tests (data not shown). We compared our Epifactor genes that showed up in in situ screening with transcriptionally downregulated genes ($p_{ajd} < 0.05$) 1 h after activated in DG under different kinds of stimuli, including kainic acid injection (KA)[33], novel environment (NE)[2], and electroconvulsive stimulation (ESC)[19] (Fig. 1e, Supplementary Data 3). We found that *Kdm4a* is the only gene that was overlapping in all these conditions. To validate this finding, we designed shRNA targeting *Kdm4a* to reduce its expression in DG neurons (Supplementary Fig. 5a-e). Consistently, in the contextual fear conditioning task, reducing *Kdm4a* expression in a subset of granule cells increased the preference of engaged activity in those cells (Fig. 1f-j and Supplementary Fig. 1g-i). Among all the Egr1-positive neurons, 20% of the cells were infected with virus expressing Kdm4a shRNA, but this number fell to 11% with the control virus (Fig. 1k). We found that reducing Kdm4a expression did not significantly increase the total number of engram cells in DG for the contextual fear conditioning task, suggesting it engaged the allocation of engrams but did not increase the population size of engram cells (Fig. 1j). In addition to use Egr1 as marker for neuronal activity, we also performed immunostaining on Fos, another marker of neuronal activity and obtained a similar result as Egr1 (Supplementary Fig. 5f, g). These data indicated that Kdm4a is a negative regulator of memory engram allocation in DG. At the same time, knockdown of Kdm4a did not increase the number of activated neurons in DG, but placed the neuron in a position of priming state to be allocated for upcoming memories.

### Neuronal activation induces immediate reduction of kdm4a expression

Next, we asked if Kdm4a expression is regulated by physiological conditions, specifically if it is regulated by neural activity. In cultured neurons (DIV 5), we found that depolarization of cultured neurons with 55 mM KCl for 1 h significantly decreases the transcription level of Kdm4a (Fig. 2a). In mouse brain, after kainic acid (KA) treatment (25 mg/kg, i.p.), which induced seizure immediately[33], hippocampal DG neurons were activated as shown by increased *Fos* and *Egr1* mRNA levels. Significant reduction of Kdm4a expression was detected in the dentate gyrus (Fig. 2b). Similarly, Pentylenetetrazole (PTZ) treatment (50 mg/kg, i.p.), a GABA receptor antagonist[2], also reduced Kdm4a expression in DG 1 h after injection and then returned to the basal levels within 6 h (Supplementary Fig. 6a-e). Besides transcription, reduction of Kdm4a protein in the nucleus was significant after depolarization in cultured neurons (Fig. 2c-e). Furthermore, fear-conditioning-activated neurons in DG showed reduced Kdm4a transcription. In *Egr1-EGFP* mice, we collected the EGFP⁺ cells (activated neurons) and EGFP⁻ cells (silent neurons) from the DG at 1.5 h after contextual fear conditioning (Supplementary Fig. 6f-j). While immediate early genes, such as *Egr1*, increased in the activated population, comparing to the EGFP⁻ population, *Kdm4a* transcription level showed significant reduction (Fig. 2f).

To further confirm that Kdm4a is down-regulated in the fear memory-activated neuron population, we isolated nuclei from mouse DG after contextual fear conditioning training by fluorescence-activated nuclei sorting (FANS) (Supplementary Fig. 7a-c). Individual nuclei were isolated from DG using Dounce homogenization. To identify the fear memory-activated DG GCs, nuclei were co-immunostained with NeuN and Fos antibodies. DG GC nuclei were identified using Hoechst-33342 via flow sorting. We found that contextual fear activated neurons (NeuN⁺Fos⁺ population) significantly

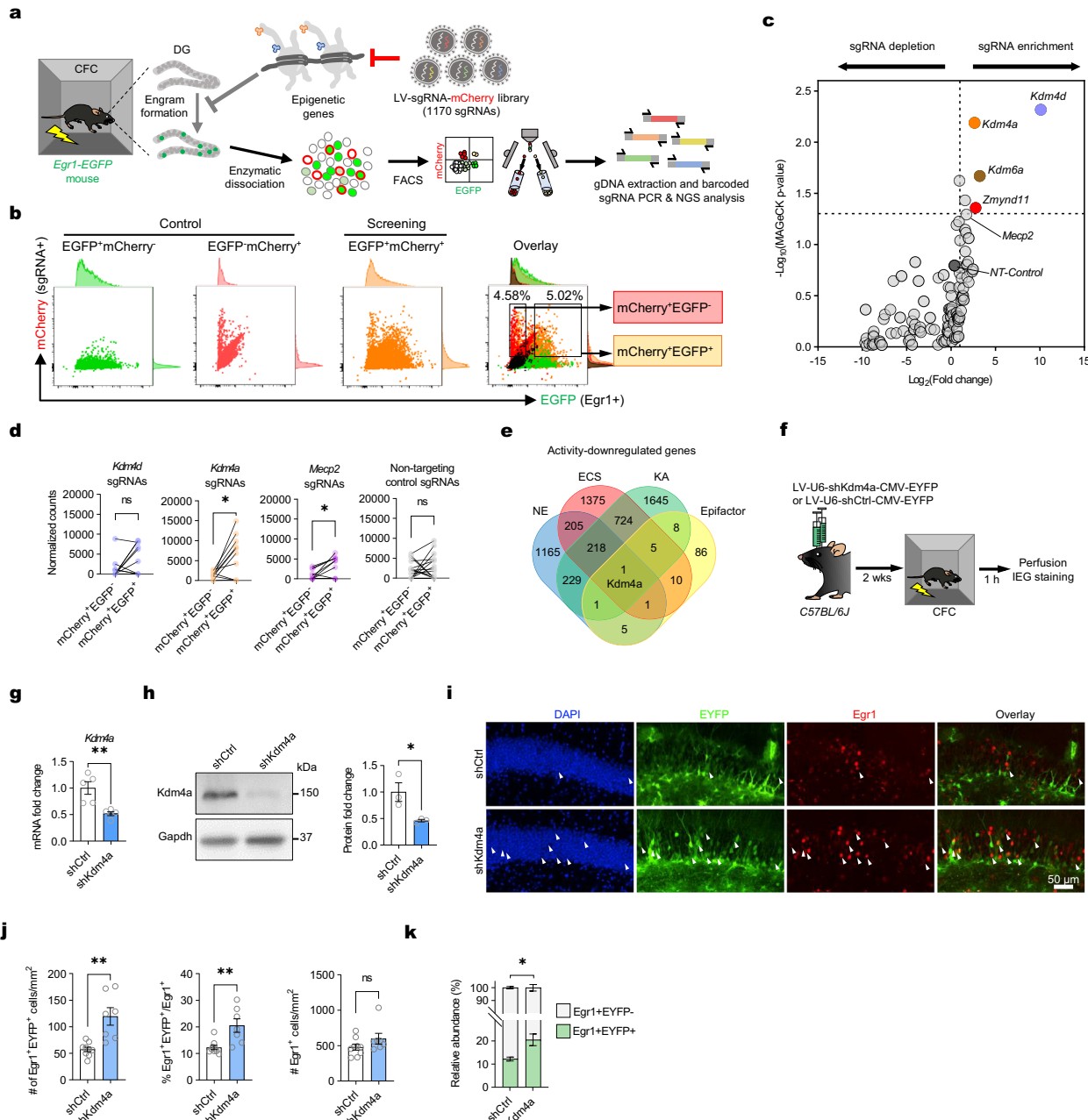

**Fig. 1 | in vivo CRISPR-ko screen identified Kdm4a as a negative regulator for memory engram formation in hippocampus. a** Schematic diagram of in vivo CRISPR-ko screen strategy. **b** Gating strategy for sorting of mCherry⁺EGFP⁻ and mCherry⁺EGFP⁺ cells. **c** Differentially enriched genes in mCherry⁺EGFP⁺ cells compared to mCherry⁺EGFP⁻ cells. Cut-off: $Log_2(FC) > 1$, $p < 0.05$, analyzed by MAGeCK. **d** Enrichment of sgRNAs of candidate genes and non-targeting control (NT-Control) sgRNAs in mCherry⁺EGFP⁻ and mCherry⁺EGFP⁺ cell populations. Two-tailed paired t-test, *Kdm4d* sgRNAs: n = 10, $t_9 = 1.989$, $p = 0.0779$; *Kdm4a* sgRNAs: n = 10, $t_9 = 3.139$, * $p = 0.0119$; *Mecp2* sgRNAs: n = 10, $t_9 = 3.218$, * $p = 0.0105$; *NT-Control* sgRNAs: n = 15, $t_{14} = 0.7639$, $p = 0.4576$. Data are presented as mean ± s.e.m. **e** Venn diagram of overlapping of epifactor-targeted CRISPR-library and genes downregulated in three transcriptome data from in vivo activated DG neurons (KA, kainic acid[33]; NE, novelty exploration[2]; ECS, electroconvulsive stimulation[19];). **f** Schematics of delivering shKdm4a or shCtrl lentivirus into the DG of *C57BL/6J* mice. **g** Kdm4a shRNA significantly knockdown the transcription of *Kdm4a* in DG neurons in vivo. shKdm4a n = 5, shCtrl n = 5. Two-tailed unpaired t-test, $t_8 = 3.881$, ** $p = 0.0047$. Data are presented as mean ± s.e.m. **h** Kdm4a shRNA significantly knockdown the

protein expression level of Kdm4a in DG neurons in vivo. shKdm4a n = 3, shCtrl n = 3. Two-tailed unpaired t-test, $t_4 = 3.055$, * $p = 0.0379$. Data are presented as mean ± s.e.m. **i,** Immunofluorescence staining of DG GCs infected with shKdm4a or shCtrl lentivirus in mouse DG. Arrows pointed at Egr1⁺EYFP⁺ double positive cells. Scale bar, 50 μm. **j** (Left) The number of Egr1⁺EYFP⁺ double-positive cells in shKdm4a mice is greater than that in the shCtrl mice. shCtrl, slices = 9; shKdm4a, slices = 7. Two-tailed unpaired t-test, $t_{14} = 4.103$, ** $p = 0.0011$. (Middle) The percentage of Egr1⁺EGFP⁺ cells/Egr1⁺ cells in the shKdm4a group was significantly higher than that in the shCtrl group. Two-tailed unpaired t-test, $t_{14} = 3.435$, ** $p = 0.0040$. (Right) The total number of Egr1⁺ cells in the shKdm4a group did not change significantly compared with the shCtrl group. Two-tailed unpaired t-test, $t_{14} = 1.476$, $p = 0.1620$. Data are presented as mean ± s.e.m. **k** The shKdm4a mice showed an increase in the fraction of Egr1⁺EYFP⁺ cells (green bars) and a reduction in the fraction of Egr1⁺EYFP⁻ cells (white bars). shCtrl, slices = 9; shKdm4a, slices = 7. One-way ANOVA followed by Bonferroni test, $F_{3,28} = 557.2$, $p < 0.0001$. * $p = 0.0112$. Data are presented as mean ± s.e.m.

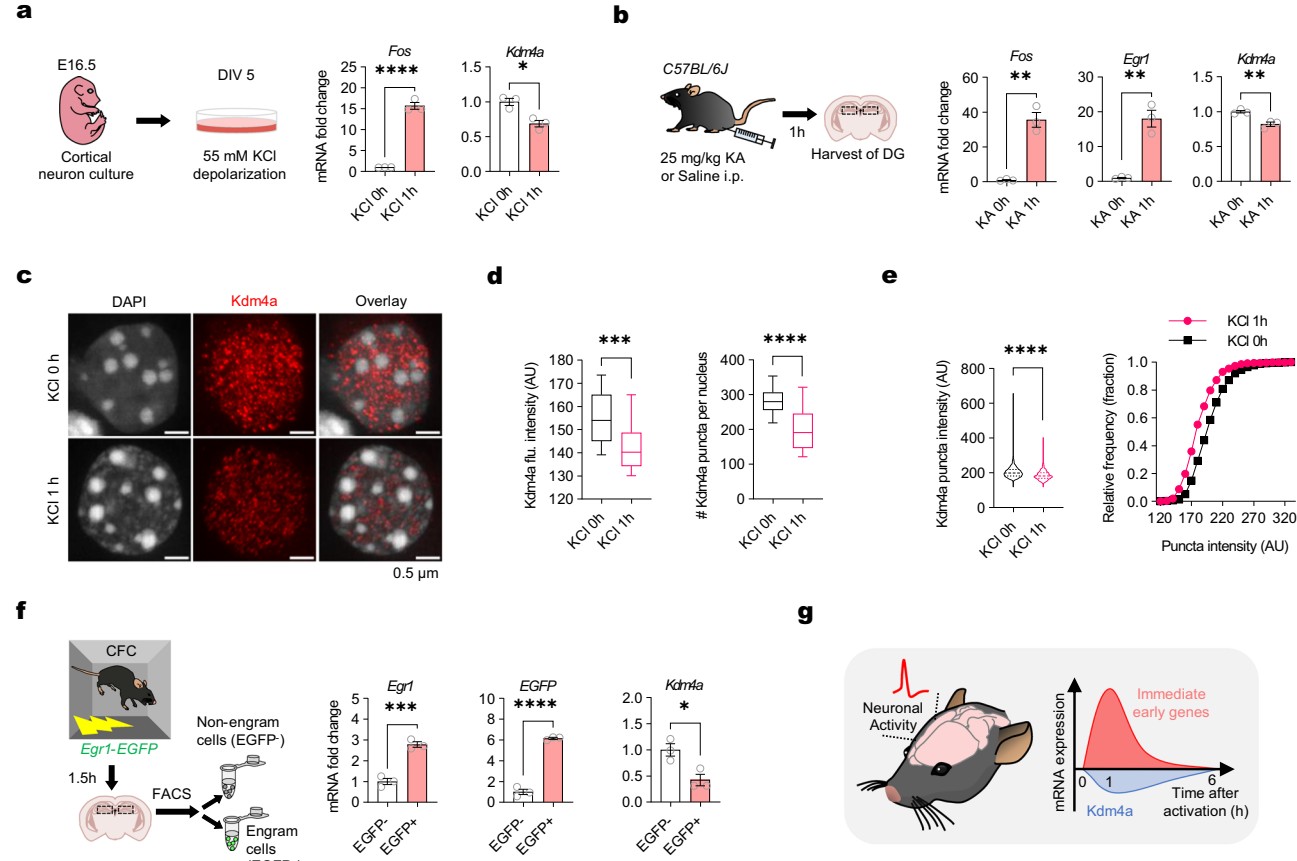

**Fig. 2 | Neuronal activation induces rapid reduction of kdm4a expression.**
**a** Schematic diagram of cultured neurons stimulated by KCl depolarization. KCl 0 h n = 3, KCl 1 h n = 3. Two-tailed unpaired t-test, *Fos*: $t_4$ = 17.84, **** $p$ < 0.0001; *Kdm4a*: $t_4$ = 4.532, * $p$ = 0.0106. Data are presented as mean ± s.e.m. **b** Schematic diagram of in vivo activation of DG using kainic acid (KA). KA 0 h n = 3, KA 1 h n = 3. Two-tailed unpaired t-test, *Fos*: $t_4$ = 8.200, ** $p$ = 0.0012; *Egr1*: $t_4$ = 7.078, ** $p$ = 0.0021; *Kdm4a*: $t_4$ = 4.980, ** $p$ = 0.0076. Data are presented as mean ± s.e.m. **c** The distribution of Kdm4a signal in the nucleus of NeuN⁺ cultured neurons (DIV14) treated with KCl 0 h or 1 h. Scale bar, 0.5 μm. **d** (Left) The Kdm4a intensity in the nucleus of NeuN⁺ neurons significantly decreased after KCl depolarization. KCl 0 h n = 15; KCl 1 h n = 26. Two-tailed unpaired t-test, $t_{39}$ = 4.102, *** $p$ = 0.0002. (Right) The number of Kdm4a puncta decreased significantly in KCl depolarized neurons. KCl 0 h n = 15;

KCl 1 h n = 22. Two-tailed unpaired t-test, $t_{35}$ = 5.210, **** $p$ < 0.0001. Box plot indicates the mean, interquartile range and the minimum and maximum. Arbitrary unit, AU. **e** The intensity of Kdm4a puncta decreased significantly after KCl depolarization. KCl 0 h n = 3, puncta = 4247; KCl 1 h n = 3, puncta = 2946. Two-tailed unpaired t-test, $t_{7191}$ = 24.08, **** $p$ < 0.0001. Violin plot indicates the mean, interquartile range and the minimum and maximum. Right: Relative frequency distribution of Kdm4a puncta intensity. **f** Schematic diagram of harvest of fear memory-related DG GCs in vivo. EGFP⁻ n = 3, EGFP⁺ n = 3. Two-tailed unpaired t-test, *Egr1*: $t_4$ = 8.902, *** $p$ = 0.0009; *EGFP*: $t_4$ = 18.84, **** $p$ < 0.0001; *Kdm4a*: $t_4$ = 3.567, * $p$ = 0.0234. Data are presented as mean ± s.e.m. **g** Schematic illustrating neuronal activation induced rapid downregulation of *Kdm4a* expression.

reduced the expression of Kdm4a compared to those silent neurons (NeuN⁺Fos⁻ population) (Supplementary Fig. 7d). Therefore, neuronal activity induces immediate reduction of Kdm4a expression in single DG neuron.

In addition to the neuronal activity-dependent expression regulation, Kdm4a is also regulated during the maturation of central nerve system. According to previous transcriptomic studies, Kdm4a expression in the brain decreases rapidly from new born to adult[34,35]. We also confirmed that Kdm4a expression in the hippocampus is significantly and gradually reduced in the postnatal stage (Supplementary Fig. 6k, l). Our study further demonstrated that Kdm4a expression in adult mouse is downregulated immediately after activation (Fig. 2g), potentially involved in the regulation of memory allocation.

### *Trpm7* is a downstream target of Kdm4a via epigenetic suppression of its transcription

To dissect the molecular mechanisms underlying kdm4a-mediated memory allocation, we examined *Kdm4a* knockdown-induced transcriptional changes through RNA sequencing (RNA-seq) by comparing the neurons with Kdm4a shRNA to the neurons with control shRNA in dentate gyrus of naïve mice (Fig. 3a and Supplementary Fig. 8). We

identified 650 downregulated genes and 275 upregulated genes in *Kdm4a*-shRNA knockdown comparing to control shRNA-treated mice (Supplementary Data 4). We did not identify key regulators of memory allocation[10,11], such as *Creb1* or *Ccr5* in this group. Gene ontology (GO) analysis revealed that the upregulated genes in *Kdm4a*-knockdown neurons were enriched in the regulation of the mitogen-activated protein kinase (MAPK) cascade, suggesting that Kdm4a may function as a negative regulator for response to neuronal stimuli (Supplementary Fig. 8d-e). Focusing on synaptic functions, we found that a melastatin-type transient receptor potential (TRPM) ion channel gene[36], *Trpm7*, was upregulated in *Kdm4a*-knockdown neurons. Trpm7 mediates calcium ion influx in the presynaptic membrane and its expression modulates learning[37,38]. RT-qPCR analysis confirmed that Kdm4a deficiency significantly increased the mRNA levels of *Trpm7* in DG granule cells (Fig. 3b and Supplementary Fig. 8b, c).

Next, we investigated how Kdm4a regulates the expression of *Trpm7*. The JmjC domain of Kdm4a catalyzes demethylation at histone H3K36me3 and H3K9me3[39,40]. We examined the trimethylation level of histone H3K36 and H3K9 and found that the removal of Kdm4a caused an increase of H3K36me3, but not H3K9me3, in the DG GCs (Fig. 3c). H3K36me3 plays a crucial role in transcriptional

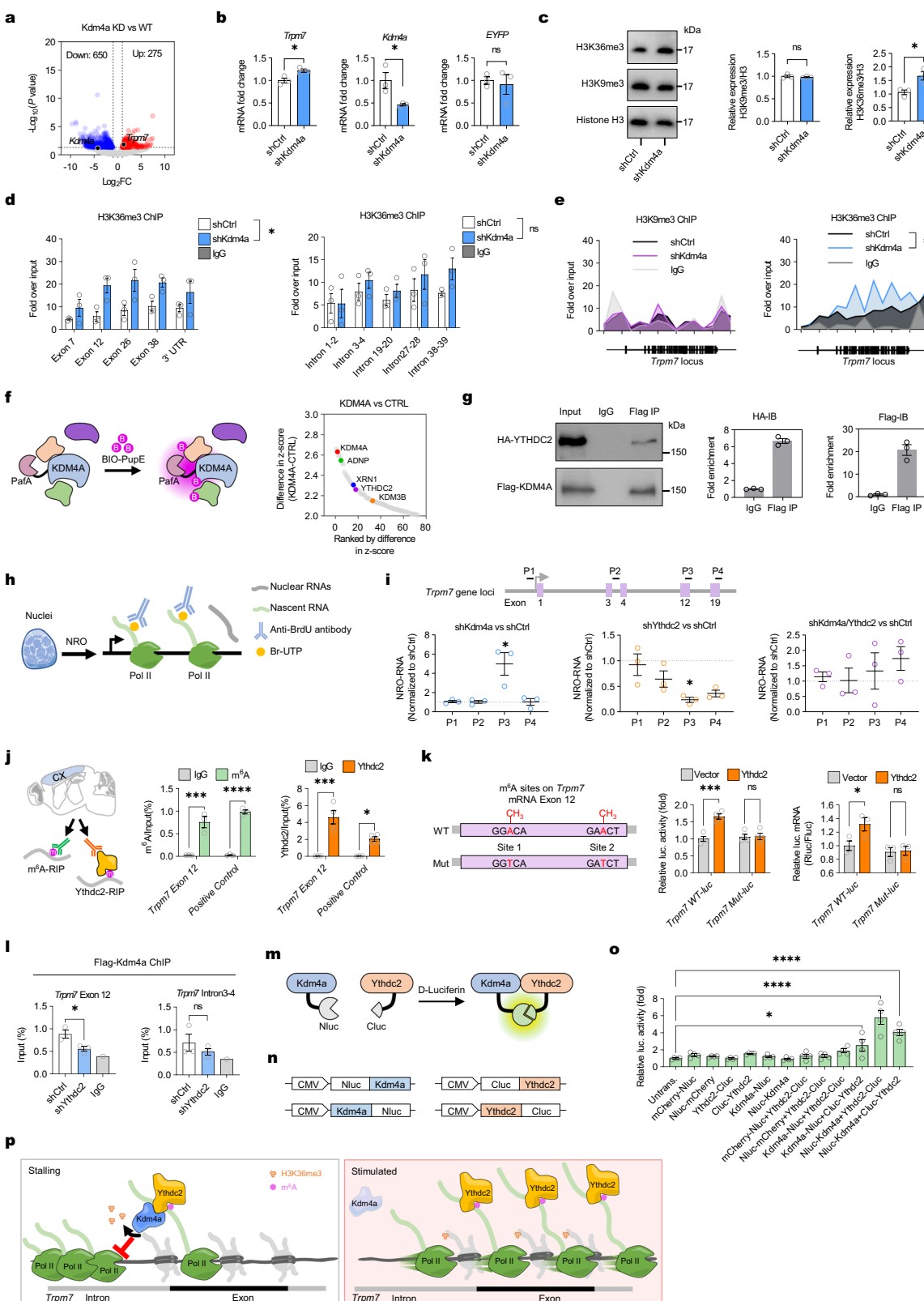

activation[41], we confirmed that H3K36me3 modifications were increased on the *Trpm7* gene loci in the Kdm4a knockdown neurons. This suggests that the enrichment of H3K36me3 is correlated with increased expression of *Trpm7* mRNA expression levels after Kdm4a knockdown (Fig. 3d, e and Supplementary Fig. 8f). Surprisingly, the H3K36me3 was not evenly affected along the gene body,

but showed a preference in the exon regions, rather than the intron regions of *Trpm7* gene loci (Fig. 3d). The preferential modification of H3K36me3 in exon regions by Kdm4a and the underlying mechanism of Kdm4a recruitment to the *Trpm7* gene loci remained unknown. To investigate this point, we further identified the crucial regulators that are involved in this molecular process.

**Fig. 3 | Kdm4a regulates Trpm7 expression via RNA Pol II pausing at exonal region. a** Differentially expressed genes were selected according to the conditions of $p < 0.05$ and |log$_2$(fold change)| > 1. Two-sided test, Benjamini-Hochberg multiple test correction. **b** Knockdown of *Kdm4a* in DG GCs resulted in upregulation of the transcription level of the cation channel Trpm7. shCtrl n = 3, shKdm4a n = 3. Two-tailed unpaired t-test, *Trpm7*: $t_4 = 3.299$, * $p = 0.0300$; *Kdm4a*: $t_4 = 3.055$, * $p = 0.0379$; *EYFP*: $t_4 = 0.3798$, $p = 0.7234$. Data are presented as mean ± s.e.m. **c** Knockdown of *Kdm4a* in DG GCs increased the level of histone H3K36me3 modification. shKdm4a n = 3, shCtrl n = 3. Two-tailed unpaired t-test, H3K9me3: $t_4 = 0.6265$, $p = 0.5649$; H3K36me3: $t_4 = 3.473$, * $p = 0.0255$. Data are presented as mean ± s.e.m. **d** (Left) Knockdown of *Kdm4a* resulted in a significant increase in the level of histone H3K36me3 modification on the *Trpm7* exonic regions. shKdm4a n = 3, shCtrl n = 3. Two-way ANOVA followed by Sidak test, $F_{1,4} = 16.0$, * $p = 0.0161$. (Right) Knockdown of *Kdm4a* did not significantly increase the level of histone H3K36me3 modification in the *Trpm7* intronic regions. shKdm4a n = 3, shCtrl n = 3. Two-way ANOVA followed by Sidak test, $F_{1,4} = 1.367$, $p = 0.3072$. **e** (Left) Knockdown of *Kdm4a* did not affect the level of histone H3K9me3 modification on the *Trpm7* gene loci. shKdm4a n = 3, shCtrl n = 3. Two-way ANOVA followed by Sidak test, $F_{1,4} = 0.2911$, $p = 0.6182$. Data are presented as mean. (Right) Knockdown of *Kdm4a* leads to an increase in the level of histone H3K36me3 modification on the *Trpm7* gene. shKdm4a n = 3, shCtrl n = 3. Two-way ANOVA followed by Sidak test, $F_{1,4} = 10.44$, * $p = 0.0319$. **f** Mass spectrometry analysis of KDM4A PUP-IT proximately labeled proteins. (Left) Schematic diagram of the PUP-IT proximity labeling system. (Right) Plot showing enriched proteins (z-score > 2.0) that were proximal to the KDM4A in living cells. **g** (Left) The Co-IP results of KDM4A and YTHDC2 indicate that KDM4A interacts with YTHDC2. Co-IP experiments were repeated for 3 times. Data are presented as mean ± s.e.m. **h** Schematic representation of in vitro nuclear real-time transcription assay (or nuclear run-on assay, NRO). **i**, NRO-qPCR results for nascent RNAs. (Up) NRO-qPCR primers designed for the mouse *Trpm7* gene loci. Primers P1 to P4 target 5'UTR, Intron 3-4, Exon 12 and Exon 19 on the *Trpm7* gene loci respectively. (Down) Knockdown of *Kdm4a* increased the transcription velocity of nascent RNA in the *Trpm7* E12 region. n = 3, unpaired two-tailed t-test, P1: $t_4 = 0.6641$, $p = 0.04680$; P2: $t_4 = 0.1909$, $p = 0.8579$; P3: $t_4 = 3.345$, * $p = 0.0287$; P4: $t_4 = 0.09449$, $p = 0.9293$. Knockdown of *Ythdc2* reduces the transcription velocity of nascent RNA in the Trpm7 E12 region. n = 3, unpaired two-tailed t-test, P1: $t_4 = 0.1281$, $p = 0.9043$; P2: $t_4 = 1.287$, $p = 0.2677$; P3: $t_4 = 3.065$, * $p = 0.0375$; P4: $t_4 = 1.64$, $p = 0.1763$. Kdm4a/Ythdc2 double knockdown has no significant effect on the transcription velocity of nascent RNA in the *Trpm7* E12 region. n = 3, unpaired two-tailed t-test, P1: $t_4 = 0.3208$, $p = 0.7644$; P2: $t_4 = 0.03464$, $p = 0.974$; P3: $t_4 = 0.4775$, $p = 0.658$; P4: $t_4 = 1.353$, $p = 0.2474$. Data are presented as mean ± s.e.m. **j** Schematic diagram of RNA immunoprecipitation (RIP) experiments. n = 3 per group, Two-way ANOVA test followed by Sidak test, *** $p = 0.0001$, **** $p < 0.0001$. Ythdc2-RIP-qPCR results show that neuronal Trpm7 mRNA E12 can bind Ythdc2 protein. n = 3, Two-way ANOVA followed by Sidak test, * $p = 0.0309$, *** $p = 0.0001$. Data are presented as mean ± s.e.m. **k** Illustration of the Exon 12 sequence of *Trpm7* and its predicted m$^6$A sites. n = 4 per group, Two-way ANOVA followed by Sidak test, *** $p = 0.0002$. Overexpression of Ythdc2 can significantly increase the mRNA expression level of Rluc. *Trpm7* E12 Mutant results in disruption of Ythdc2 binding to RNA. n = 4, Two-way ANOVA followed by Sidak test, * $p = 0.0395$. Data are presented as mean ± s.e.m. **l**, ChIP-qPCR assay for Flag-Kdm4a at the exonic and intronic regions of gene *Trpm7* in WT or Ythdc2 KD cells. Exon12: n = 3, two-tailed unpaired t-test, $t_4 = 3.313$, * $p = 0.0296$; Intron 3-4: n = 3, $t_4 = 0.9444$, ns $p = 0.3984$. Data are presented as mean ± s.e.m. **m** Schematics of the split luciferase complementation assay. **n**, Schematic illustrations of the fused protein variants. **o**, Relative luminescence intensity of each pair of interacting partners. n = 4 per group, One-way ANOVA followed by Bonferroni test, $F_{12,39} = 17.6$, $p < 0.0001$. * $p = 0.0366$, **** $p < 0.0001$. Data are presented as mean ± s.e.m. **p** Schematic diagram of Ythdc2 binds to m$^6$A sites in the exonic region of *Trpm7* mRNA to recruit the transcription repressor Kdm4a to clear the histone H3K36me3 methylation modifications, thereby reducing the transcription velocity of the exon region of the Trpm7 gene.

## Kdm4a is recruited to the genomic loci via Ythdc2 to allow burst expression of *Trpm7*

To identify co-factors recruiting Kdm4a to *Trpm7* gene loci, we performed biotin-mediated proximity labeling[42] to map the Kdm4a-associated proteome in living cells and conducted mass spectrum analysis (Fig. 3f and Supplementary Fig. 9a-e). Among all the candidates interacting with Kdm4a, RNA m$^6$A reader YTH domain containing 2 (Ythdc2)[43] emerged as a key factor in recruiting Kdm4a (Supplementary Data 5). Firstly, Co-immunoprecipitation and western blot confirmed the binding between Kdm4a and Ythdc2 in HEK293T cells (Fig. 3g). Secondly, the expression of YTHDC2 showed a consistent correlation with the expression of KDM4A in various tissues (n = 54, $r = 0.84$, $p < 0.0001$) (Supplementary Fig. 8f, g). Thirdly, immunostaining of the nucleus indicated that Kdm4a and Ythdc2 were colocalized in some punctas (Supplementary Fig. 8h).

As H3K36m3 modifications spread throughout the gene body regions but not the promoter region[44], we next investigated the role of Kdm4a-Ythdc2 complexes in transcription of nascent RNAs by measuring the transcription speed through an in vitro nuclear run-on assay (Fig. 3h). Kdm4a and Ythdc2 were knocked down separately or simultaneously in Neuro-2a cells using shRNAs (Supplementary Fig. 10a, b). Then, the nuclei were purified and Br-UTP was added to label the newly synthesized nascent RNAs. Interestingly, Kdm4a did not affect nascent RNA synthesis in the promoter region or the inronic regions between exon 3 and exon 4. In contrast, knockdown of Kdm4a did produce more nascent RNAs, containing exon 12 (Fig. 3i). As those nascent RNAs were on the same gene, the difference between them indicated the pausing of newly synthesized RNAs accumulated before exon 12 in the presence of Kdm4a. After the removal of the blockage (Kdm4a), increased production of nascent RNAs containing exon 12 was observed in the nuclear run-on assay.

Furthermore, as Ythdc2 prevents RNA decay[45], knockdown of Ythdc2 induced a decreased amount of synthesized nascent RNAs in the gene body but not the promoter region of *Trpm7* (Fig. 3i).

Interestingly, the exon 12 region showed the most significant decrease of BrUTP-containing nascent RNAs, suggesting that Ythdc2 might protect those paused nascent RNAs accumulated before exon 12. Therefore, when the RNAs were synthesized at the *Trpm7* loci, Kdm4a induced the pausing of RNA synthesis before exon 12 to accumulate a large number of half-synthesized RNAs, and Ythdc2 further protected those paused RNAs from decay. In Kdm4a/Ythdc2 double-knockdown cells, nascent RNAs containing exon 12 did not increase significantly, suggesting that the pausing effect induced by Kdm4a around exon 12 of *Trpm7* gene loci is dependent on Ythdc2 (Fig. 3i). Ythdc2 predominantly binds to the m$^6$A sites in the coding sequencing (CDS) region of RNA[46], we identified two potential m$^6$A sites (1634A, 1713A) in the *Trpm7* mRNA exon 12 for Ythdc2 binding through SRAMP prediction[47] (Supplementary Fig. 10c). By conducting m$^6$A-RNA Immunoprecipitation (RIP) and Ythdc2-RIP, we found that *Trpm7* mRNA is modified by m$^6$A and Ythdc2 directly binds *Trpm7* mRNA in mouse brain (Fig. 3j). To further demonstrate the binding of Ythdc2 to *Trpm7* mRNA is mediated by m$^6$A, we mutated two possible m$^6$A sites on *Trpm7* exon 12 sequence (from 1591 to 1725) and performed dual luciferase reporter assay (Supplementary Fig. 10d, e). Mutations in the m$^6$A sites in exon12 abolished the Ythdc2 overexpression-induced mRNA protection effect (Fig. 3k and Supplementary Fig. 10f).

The Ythdc2-dependent recruitment of Kdm4a to the genomic loci was further confirmed by ChIP assay (Fig. 3l) around the exon 12 region. Furthermore, the interaction between Ythdc2 and Kdm4a was further validated via a split luciferase-based protein-fragment complementation assay (PCA)[48]. A significant luminescence signal was detected when Nluc-Kdm4a (Nluc, amino acid 2-416, N-terminal of firefly luciferase protein was attached to Kdm4a in N terminal) was coexpressed with Ythdc2-Cluc (Cluc, amino acid 389-550, C-terminal of firefly luciferase protein was attached to Ythdc2 in C terminal), confirming that Kdm4a interacts with Ythdc2 in vivo (Fig. 3m−o).

Taken together, these evidence suggest that Kdm4a is recruited by Ythdc2 to the exon 12 of *Trpm7* gene to remove H3K36me3 and

cause a temporal stalling effect on the RNA synthesized before exon 12, leading to the accumulation of a large pool of half-synthesized RNAs. Upon stimulation and breakdown of Kdm4a, a large amount of *Trpm7* RNAs are synthesized rapidly. Ythdc2 recognizes m⁶A sites to accumulate in exon 12 of *Trpm7* genomic loci. At the same time, Ythdc2 protects those nascent RNAs from decay to increase the burst transcription of the *Trpm7* gene (Fig. 3p).

## Kdm4a and Trpm7 significantly regulate the size of mossy fiber boutons

To understand how the Kdm4a regulates allocation of engrams, we tested whether the Kdm4a regulates the maturation of memory circuits, especially the synaptic structures, as Trpm7 regulates calcium influx in presynaptic structures[37]. By using lentiviral shRNA knockdown (LV-U6-shKdm4a-CMV-EYFP), we examined the role of Kdm4a in regulating synapse morphology in granule cells of the DG (Fig. 4a). After analyzing the morphology of dendrite spines in the DG molecular layer (ML), we did not find any alternations in the spine density, spine head width, or the percentage of matured mushroom synapses (Fig. 4b, c). Instead, we found alternations in the axonal regions of DG granule cells, specifically mossy fiber terminals, which showed significant changes in the stratum lucidum (SL) layer of CA3 (Fig. 4d). Knockdown of Kdm4a in the DG significantly increased the size of mossy fiber boutons (MFBs) (Fig. 4e, f). Consistently, after inducing high expression levels of *Trpm7* via CRISPRa-based transcription activation in the DG (Fig. 4g and Supplementary Fig. 11a-d), we found that the size of MFBs was significantly increased (Fig. 4h, i and Supplementary Fig. 11e). Thus, surprisingly, our data suggest that Kdm4a does not potentiate the inputs but regulates presynaptic function that controls the output of the potential engram cells. Although enlargement of MFBs might not directly affect activity in dentate gyrus, these observations suggest that Trpm7 and Kdm4a regulate presynaptic function and might regulate the maturation of dentate gyrus circuits for memory allocation. In addition, as Trpm7 has been reported to be essential for synaptic plasticity in early postnatal conditions[37,38], our results further suggest that Kdm4a and Trpm7, one of the downstream targets of Kdm4a, both regulate the maturation of the granule cell circuits, potentially contributing to the allocation of memory engrams.

## Reducing Kdm4a expression in DG neurons facilitated the decoupling of contextual memories adjacent in time

Finally, we asked if increased engram allocation could facilitate the separation of adjacent memories. We stereotaxically delivered CRISPRko-based Kdm4a-knockout lentivirus or control virus into the mouse DG (Fig. 5a, b and Supplementary Fig. 12a-h). *Kdm4a^DG-KO* mice showed normal locomotor activity in the open field test (Fig. 5c). They showed similar performance in memory encoding in a contextual fear conditioning test (Fig. 5d). In the fear extinction test, *Kdm4a^DG-KO* mice also showed similar performance as the control group (Fig. 5e). Thus, manipulating the engram allocation in the DG did not significantly alter the encoding, retrieval and extinction of the contextual fear memory in those *Kdm4a^DG-KO* mice.

Next, we put those mice into the contextual fear discrimination test (CFD). In this test (Fig. 5f), mice were placed in two different contexts: context A and B. The mice received a foot-shock in the context A but not the context B throughout the test for 4 days. Mice were placed in the two contexts sequentially with 2 h interval. Although the foot-shock was only associated with context A, due to the adjacent in time, mice showed a significant amount of freezing in context B. Interestingly, *Kdm4a^DG-KO* mice showed normal fear response in context A, but significant lower freezing levels in context B, when compared with the control group (Fig. 5g). *Kdm4a^DG-KO* mice achieved high discrimination score in this test, indicating that virally introduce a subpopulation of Kdm4a-downregulated dentate neurons separate distinct contextual memories close in time. Similar effects of higher discrimination between

context A and context B in the Kdm4a cKO mice were also detected in an immediate shock task (Supplementary Fig. 13e). When mice were immediate shocked in context A (10 s habituation, 2 s footshock, 30 s delay), the tests at 5 h after the shock showed higher freezing rate in context B in the control group. However, Kdm4a cKO mice showed significantly low level of freezing in context B, compared to the immediate freezing in context A(Supplementary Fig. 13f).

To further extend this discovery, we asked if contextual discrimination is affected by artificially activating DG neurons, which are capable of inducing an immediate reduction of Kdm4a expression (Fig. 2), before the events. To this end, we delivered adeno-associated virus (AAV) expressing Gq-coupled designer receptor exclusively activated by designer drugs (hM3Dq) into the mouse DG area (Fig. 5h and Supplementary Fig. 13a, b). The hM3Dq-based artificial activation increased the activity of neurons in DG, as indicated by cFos, at 1 h after injection of clozapine-*N*-oxide (CNO) (2 mg/kg, i.p.) (Fig. 5i). Consistently, expression of mRNA in the dentate gyrus showed increased signal of *Fos* and *Egr1* gene and reduced signal of *Kdm4a* gene after CNO induced neuronal activation (Supplementary Fig. 13c). We performed a similar context discrimination task when two contexts were placed close in time within 3 h, and DG neurons were artificially activated about 1 h before context B training. We found that compared to the control group, hM3Dq expressing mice performed better discrimination between context A and B in the presence of CNO (Fig. 5k). On day 4, when no CNO was present, the hM3Dq group showed much less freezing comparing to other groups (Fig. 5j, k). However, in the saline-injected groups, no difference was detected between the hM3Dq group and the control groups, indicating that pre-task activation was required to discriminate between the two contexts adjacent in time (Fig. 5k).

We performed the memory allocation test to determine whether artificially decreasing *Kdm4a* levels in a subset of DG neurons affects the ability to allocate contextual memories that occurred close in time. Mice injected with LV-U6-shKdm4a or control virus were subjected to a typical memory allocation test[8]. Briefly, the mice were subjected to freely explore three distinct, novel contexts which were separated by 7 days (context C and context B) or 5 h (context A and context B). Two days later, mice were placed in context A and given an immediate footshock (Fig. 5l and Supplementary Fig. 13d). Two days after the footshock, the mice were tested in either context A (shocked context), B (5 h; not shocked), or C (7 days; not shocked). As previously reported[8], the control mice (shCtrl) froze similarly in contexts A and B, exhibited memory-linking between distinct memories encoded close in time. However, the *Kdm4a*-knockdown mice displayed a significantly lower freezing in context B than that in context A (Fig. 5m), which suggests that downregulation of *Kdm4a* in DG regulates memory allocation to separate linked memories closed in time.

## Mice with Kdm4a gene knockout in brain showed unaltered memory formation and enhanced ability of context pattern separation

To eliminate the influence of viral infection, we generated a conditional knockout (cKO) mice model to induce *Kdm4a* KO in brain (Fig. 6a and Supplementary Fig. 14). Following *Nestin-Cre*-mediated recombination, the *Kdm4a* cKO mice showed significant depletion of Kdm4a RNA and protein in the cortical and hippocampal regions (Fig. 6b−e). The *Kdm4a cKO* mice showed normal locomotor activity in the open field test and unaltered anxiety levels in the elevated plus maze test (Fig. 6f, g). In the contextual fear conditioning test, there were no significant differences in memory recall between the *Kdm4a cKO* group and the control group of mice (Fig. 6h), and their sensory responses to the electrical pulse appeared to be the same (Fig. 6i), suggesting that memory formation and retrieval remained unaltered in those mice. We then evaluated the pattern separation ability of *Kdm4a cKO* mice in the same test which was performed in DG-specific Kdm4a knockout mice (Fig. 6j). Kdm4a cKO mice showed enhanced

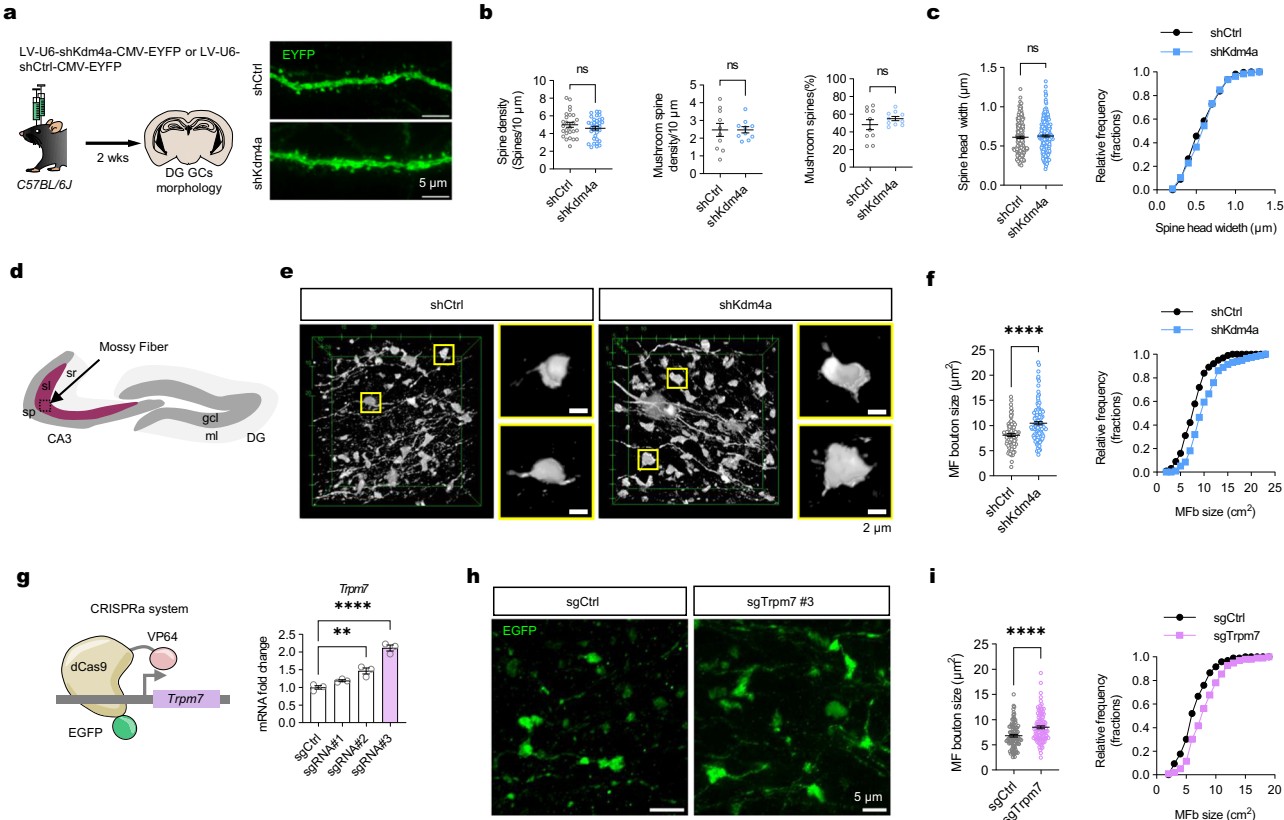

**Fig. 4 | Kdm4a and Trpm7 significantly regulate the size of mossy fiber boutons. a** Schematic of sparse labeling of the mouse DG GCs using shKdm4a or shCtrl lentivirus. Scale bar, 5 μm. **b** Dendritic spine density are not significantly different between shKdm4a (n = 3, dendrites = 34) and shCtrl (n = 3, dendrites = 26) mice. Two-tailed unpaired t-test, $t_{58} = 1.142$, $p = 0.2583$. The density of mushroom-like dendritic spines are similar between shKdm4a (n = 3, dendrites = 10) and shCtrl (n = 3, dendrites = 10). Two-tailed unpaired t-test, $t_{18} = 0.01855$, $p = 0.9854$. The proportion of mushroom-like dendritic spines are not significantly different between shKdm4a (n = 3, dendrites = 10) and shCtrl (n = 3, dendrites = 10). Two-tailed unpaired t-test, $t_{18} = 1.109$, $p = 0.2818$. Data are presented as mean ± s.e.m. **c** The width of dendritic spines are similar between shKdm4a (n = 3, spines = 265) and shCtrl (n = 3, spines = 265). Two-tailed unpaired t-test, $t_{528} = 0.9885$, $p = 0.3233$. Right: Frequency distribution plot of dendritic spine width. Data are presented as mean ± s.e.m. **d** Schematic illustration of subregions of mouse dentate gyrus and CA3. Granule cell layer: gcl, molecular layer: ml stratum pyramidalis: sp stratum

lucidum: sl stratum radiatum: sr. **e** Immunofluorescence staining of EYFP⁺ neuron axonal projections in the stratum lucidum of CA3. Scale bar, 2 μm. **f** The size of DG MFb was significantly increased in the shKdm4a mice (n = 3, boutons = 117) compared with shCtrl control mice (n = 3, boutons = 100). Two-tailed unpaired t-test, $t_{214} = 5.333$, **** $p < 0.0001$. (Right) Frequency distribution histogram of DG mossy fiber bouton size. Data are presented as mean ± s.e.m. **g** Illustration of CRISPR-a based transcription activation system for overexpression of *Trpm7*. n = 3, One-way ANOVA followed by Bonferroni test, $F_{3,\,8} = 53.28$, $p < 0.0001$. ** $p = 0.003$, **** $p < 0.0001$. Data are presented as mean ± s.e.m. **h** Immunofluorescence staining of EGFP⁺ neuron axonal projections in the stratum lucidum of CA3. Scale bar, 5 μm. **i,** The area of DG MFb in the sgTrpm7 mice (n = 3, boutons = 96) was significantly larger than that in the sgCtrl mice (n = 3, boutons = 96). Two-tailed unpaired t-test, $t_{190} = 4.280$, **** $p < 0.0001$. Right: Frequency distribution histogram of mossy fiber bouton size. Data are presented as mean ± s.e.m.

contextual discrimination ability and decoupled the contextual memories adjacent in time (Fig. 6k).

## Discussion

In this study, we developed the CRISPRko-based in situ screening system and identified an epigenetic regulator, *Kdm4a*, which is essential for engram allocation and memory separation. Interestingly, this factor is also subjected to neural activity-dependent down-regulation, implicating a potential role of pre-learning activity in preparing for memory allocation. Mechanistically, Kdm4a suppresses the expression of *Trpm7* by binding to the exonic regions and removes H3K36me3. We found Kdm4a was recruited to the *Trpm7* gene loci via interacting with Ythdc2, which reads the m⁶A sites on nascent RNAs. We proposed that Kdm4a induced stalling of nascent RNAs on *Trpm7* loci, which allows activity-induced burst expression of *Trpm7* to prepare synaptic proteins in neurons for memory encoding. Consequently, the knockdown of Kdm4a in the DG altered the size of MFBs and facilitated memory allocation, promoting pattern separation. Conditional knockout of Kdm4a in the brain decoupled contextual memories closed in time, which are normally linked together to

allocate memories in the same engram ensemble. The role of Kdm4a in memory regulation suggests a priming state in which neurons get prepared for memory allocation.

Memory engrams are neuronal ensembles that encode the memorized information. While they are activated during learning[49], it was unclear if the specific ensemble of neurons to encode the event is pre-determined by regulatory mechanisms. The memory allocation hypothesis suggests potential engram ensembles compete with each other for memory storage. Thereby, when the old ensembles are activated, they go through a 6-h time window with potentiated cell excitability to attract new memories into the old ensemble[9]. The memory allocation hypothesis suggests the overlapping of engram populations and the enhancement and linkage of distinct contextual memories adjacent in time[8]. In agreement with the memory allocation hypothesis, we found Kdm4a, an epigenetic factor, is able to modulate the selection of activated ensemble in DG during learning, promoting memory allocation. In fact, the activity history of dentate granule neurons does reshape both the transcriptional signature[6] and epigenome[4], thereby impacting the process of selecting of activated neurons and the reactivation of

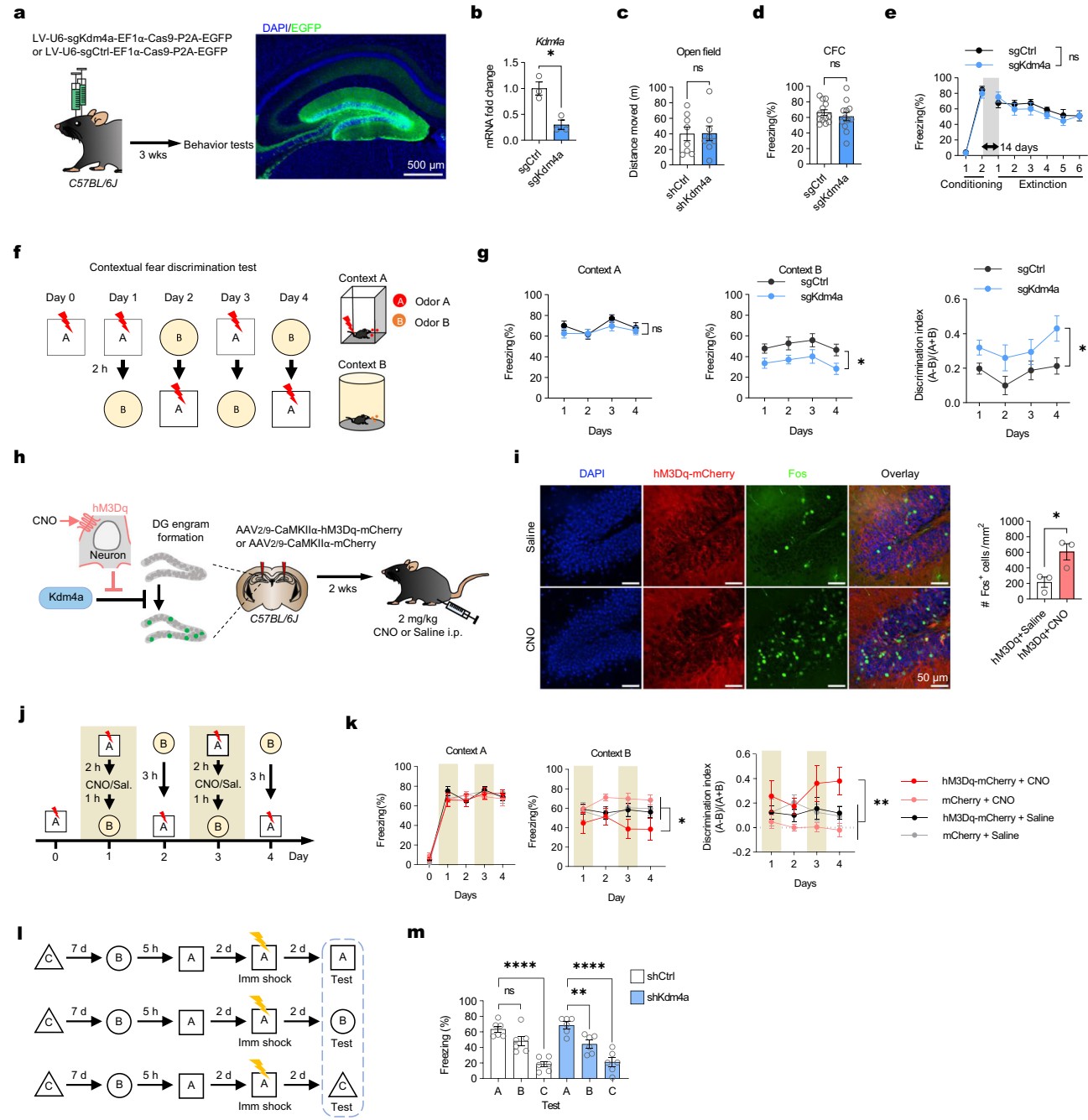

memory-related neuronal ensembles. Our studies further extended these observations and suggested that epigenetic regulation might engage a pathway for memory allocation, other than the CREB-dependent one, as the Kdm4a knockdown in a subset of dentate granule cells increased decoupling of contextual memories adjacent in time.

Kdm4a might be involved in the development stage-related regulation of memory allocation. It is known that memory precision gradually increased from juvenile to adult[49]. Kdm4a expression in the brain decreases rapidly from newborn to adult[34,35]. We found that *Kdm4a* expression in the hippocampus decreased during the maturation of the brain. Downregulation of Kdm4a could modulate synaptic connections, which adjust the maturation of the neural circuits, readily for engram allocation. Thereby, the downregulation of Kdm4a can engage a priming state of neural ensembles for memory allocation, increase pattern separation. Deficits in Kdm4a might lead to dysregulation of memory circuits, as studies have identified KDM4A in

attention-deficit/hyperactivity disorder (ADHD)[50], a prevalent neurodevelopmental disease in juveniles.

Taken together, we discovered Kdm4a as an epigenetic modifier that negatively regulates the allocation of engram cells for new memory encoding. Besides highlighting the m⁶A-mediated molecular effectors in memory regulation, this study further suggests an internally regulated mechanism that places neurons in a priming state, shedding light on the regulatory process of engram formation. It also opens a new era to studying new mechanisms underlying memory-associated disorders, such as ADHD, AD and depression.

## Methods

### Animals

Animals were raised on the Model Animal Platform at ShanghaiTech University. All animal protocols were approved by the Institutional Animal Care and Use Committee of the ShanghaiTech University. The mice were bred from *C57BL/6J* background and group-housed (3–5

**Fig. 5 | Kdm4a regulated engram formation in dentate gyrus to facilitate pattern separation task. a** Schematic of Injection of sgKdm4a or sgCtrl lentivirus into the DG of *C57BL/6J* mice. Scale bar, 500 μm. **b** RT-qPCR detection of Kdm4a expression level in DG infecting sgKdm4a versus sgCtrl lentivirus. n = 3, Two-tailed unpaired t-test, $t_4 = 4.529$, * $p = 0.0106$. Data are presented as mean ± s.e.m. **c** Knockout of Kdm4a in DG does not affect mouse locomotor activity. sgCtrl, n = 8; sgKdm4a, n = 8. Two-tailed unpaired t-test, $t_{14} = 0.04956$, $p = 0.9612$. Data are presented as mean ± s.e.m. **d** *Kdm4a^{DG-KO}* mice showed normal contextual fear learning when compared with controls. sgCtrl, n = 13; sgKdm4a, n = 12. Two-tailed unpaired t-test, $t_{23} = 0.7862$, $p = 0.4398$. Data are presented as mean ± s.e.m. **e** Knockout of Kdm4a in DG does not affect mouse fear memory extinction. sgCtrl, n = 6; sgKdm4a, n = 6. (Extinction day 1-6) Two-way ANOVA followed by Sidak test, $F_{1,10} = 0.2891$, $p = 0.6026$. Data are presented as mean ± s.e.m. **f** Illustration of contextual fear discrimination task. Mice were allowed to learn to distinguish two slightly different contexts: context A and B. **g** (Left) There was no difference in freezing levels between *Kdm4a^{DG-KO}* mice and control mice in context A. sgCtrl, n = 12, sgKdm4a, n = 9. Two-way ANOVA followed by Sidak test, $F_{1,20} = 0.5987$, $p = 0.4481$. (Middle) The freezing level of *Kdm4a^{DG-KO}* mice in context B was significantly lower than that of control mice. Two-way ANOVA followed by Sidak test, $F_{1,19} = 5.745$, * $p = 0.027$. (Right) The discrimination index of *Kdm4a^{DG-KO}* mice is higher than that of the control mice. Two-way ANOVA followed by Sidak test, $F_{1,19} = 6.792$, * $p = 0.0174$. Data are presented as mean ± s.e.m. **h** The chemogenetic virus hM3Dq-mCherry or control vector was delivered into the DG of *C57BL/6J* mice. **i**, CNO was injected 1 h before perfusion and increased Fos expression in the DG with AAV-hM3Dq. Saline, n = 3; CNO, n = 3. Two-tailed unpaired t-test, $t_4 = 3.163$, * $p = 0.0341$. Data are presented as mean ± s.e.m. **j** Schematic of the contextual fear discrimination test. CNO or saline were injected 1 h before exploring of context B each day. **k** (Left) There was no difference in freezing levels between hM3Dq+CNO mice and other control groups in context A. n = 8 per group. Two-way ANOVA followed by Dunnett test, $F_{3,28} = 0.07766$, $p = 0.9716$. (Middle) The freezing level of hM3Dq+CNO mice in context B was significantly decreased when compared with other control mice. Two-way ANOVA followed by Dunnett test, $F_{3,28} = 3.494$, * $p = 0.0285$. (Right) The discrimination index of hM3Dq+CNO mice was significantly higher than that of other control mice. Two-way ANOVA followed by Dunnett test, $F_{3,28} = 6.885$, ** $p = 0.0013$. Data are presented as mean ± s.e.m. **l** Characterization of the memory allocation test and context (Imm shock, immediate shock). **m** Kdm4a-knockdown mice froze significantly lower in context B than that in context A. shCtrl, n = 6, shKdm4a, n = 6. One-way ANOVA followed by Dunnett test, shCtrl: $F_{2,15} = 23.49$, $p < 0.0001$. ns $p = 0.0671$, **** $p < 0.0001$; shKdm4a: $F_{2,15} = 22.79$, $p < 0.0001$. ** $p = 0.0069$, **** $p < 0.0001$. Data are presented as mean ± s.e.m.

mice per cage) on a 12 h/12 h light/dark cycle with diet and water freely available. The ambient temperature was maintained at 23 °C, and humidity levels at 50%. *Tg(Egr1-EGFP)GO90Gsat* (GENSAT, strain #: 4847022) knockin mice, both male and female, aged 8–12 weeks, were bilaterally injected with LV-CRISPR library into the dorsal dentate gyrus (±1.5, −2.0, −2.0). Mice were allowed to recover from surgery for 2–3 weeks before all behavioral tasks. *B6.Cg-Tg(Nes-cre)1Kln/J* (The Jax lab, strain #: 003771) mice were crossed to *Kdm4a^{flox/flox}* mice (customed from GemPharmatech, Strain #: T052202) to generate neuron-specific Kdm4a knockout mice (*Nes-Cre^{+/Tg};Kdm4a^{f/f}*). *Nes-Cre* mice were kindly gifted by Y. Cang laboratory at ShanghaiTech University. Animals, both male and female, were randomly assigned to the experimental groups. All animal genotyping primers were outlined in the Supplementary Data 6.

## DNA constructions

Candidate sgRNAs and shRNAs sequences targeted on the genes of interest were designed using E-CRISP[51] and CRISPick (https://portals.broadinstitute.org/gpp/public/) online tools. To construct the lentiviral CRISPR plasmids, the stuffer on lenti-CRISPRko or lenti-CRISPRa backbone was replaced with annealed sgRNA oligos using restriction enzyme-based cloning. To generate lentiviral RNAi vectors, the hPGK and Puro^R fragments are on the pLKO.1 backbone (Addgene, 10878) were replaced with CMV promoter and full-length EYFP respectively using Gibson Assembly, and then the annealed shRNA oligos were inserted to the plasmid using restriction enzyme-based cloning. The sequences of CRISPR, RNAi oligonucleotides were listed in the Supplementary Data 6.

To produce the proximity labeling plasmids, KDM4A fragments were PCR amplified from the cDNA library of human 293T cell line and fused to the N-terminal of PafA with a 3xFlag-tag. KDM4A-PafA fragments were subcloned into the lentiviral plasmids using Gibson Assembly. A nuclear localization signal (NLS) was fused to the N-terminal of PafA, as a control bait. The N-terminal of Bio-PupE, the substrates of PafA, was fused to a 3x NLS tag for the localization in the nuclei. IRES fragments and EGFP or mCherry fragments were inserted into the C-terminal of KDM4A-PafA or 3x NLS-PafA using Gibson Assembly. All plasmids were verified by Sanger sequencing.

## Generation of the sgRNA library for the screen

The mouse histone methylation CRISPR sgRNA library was designed by selecting ten sgRNAs targeting each gene of interest (117 genes encoding histone methylation modifiers were obtained from the Epi-Factor Database[52,53]), along with an additional 15 non-targeting negative control sgRNAs. The oligonucleotide pool was synthesized by CustomArray, Inc. The sequences of CRISPR library oligonucleotides used in this study were outlined in Supplementary Data 1.

The pooled library sgRNA fragments, which contained homology arms, were inserted into the BsmB1-digested lentiCRISPRv2-mCherry plasmids using Gibson Assembly Kit (NEB). The ligation products were desalted and transformed into 20 μl of *DH5α* competent cells (Weidi, DE1001) using an electroporation apparatus (Bio-Rad MicroPulser). After electroporation, the cells were recovered by adding 980 μl of pre-warmed 37 °C SOC medium (Weidi, CM1014L) and incubating for 1 h at 37 °C with shaking at 250 rpm. Transformants were plated onto five 24.5 × 24.5 cm LB agar plates and incubated for 14–15 h at 30 °C. The colonies were collected and subjected to plasmid DNA extraction using an endotoxin-free plasmid DNA isolation kit.

To examine the integrity and proper representation of the sgRNAs in the library plasmids, the plasmid DNAs were used for PCR amplification of regions containing the sgRNA fragment. The PCR products were purified and determined by Next Generation Sequencing (NGS).

## Cell culture and transfection

The HEK293T (ATCC CRL-3216), HEK293FT (ATCC CRL-3249), U2-OS (ATCC HTB-96) and Neuro-2a (ATCC CCL-131) cell lines were maintained in high-glucose DMEM medium supplemented with 10% fetal bovine serum (FBS) and 1% penicillin-streptomycin at 37 °C with 5% $CO_2$. All cells were subcultured every 3–4 days when they reached 80% confluency. The transfection of HEK293T, HEK293FT cells was performed using polyethylenimine (PEI). The transfection of U2-OS and Neuro-2a cells was performed using EZ Trans Reagents (Life-iLab).

## Lentivirus production

Lentivirus was generated by transfected HEK293-FT cells in three 10-cm cell culture dishes with lentiviral transfer vectors, packaging vector psPAX2 and envelop vector pVSV-G using polyethylenimine (PEI). Media containing viral particles were harvested from cell culture at 48 and 72 h after transfection. The collected media were passed through a 0.45 μm filter to remove cell debris, and then the filtered media were centrifuged in a Beckman SW-28 rotor for 2 h at 25,000 rpm at 4 °C, and 5 ml of 20% sucrose solution was added to the bottom of the centrifuge tube before centrifugation. The concentrated lentivirus was resuspended in 20 μl ice-cold D-PBS and stored at −80 °C.

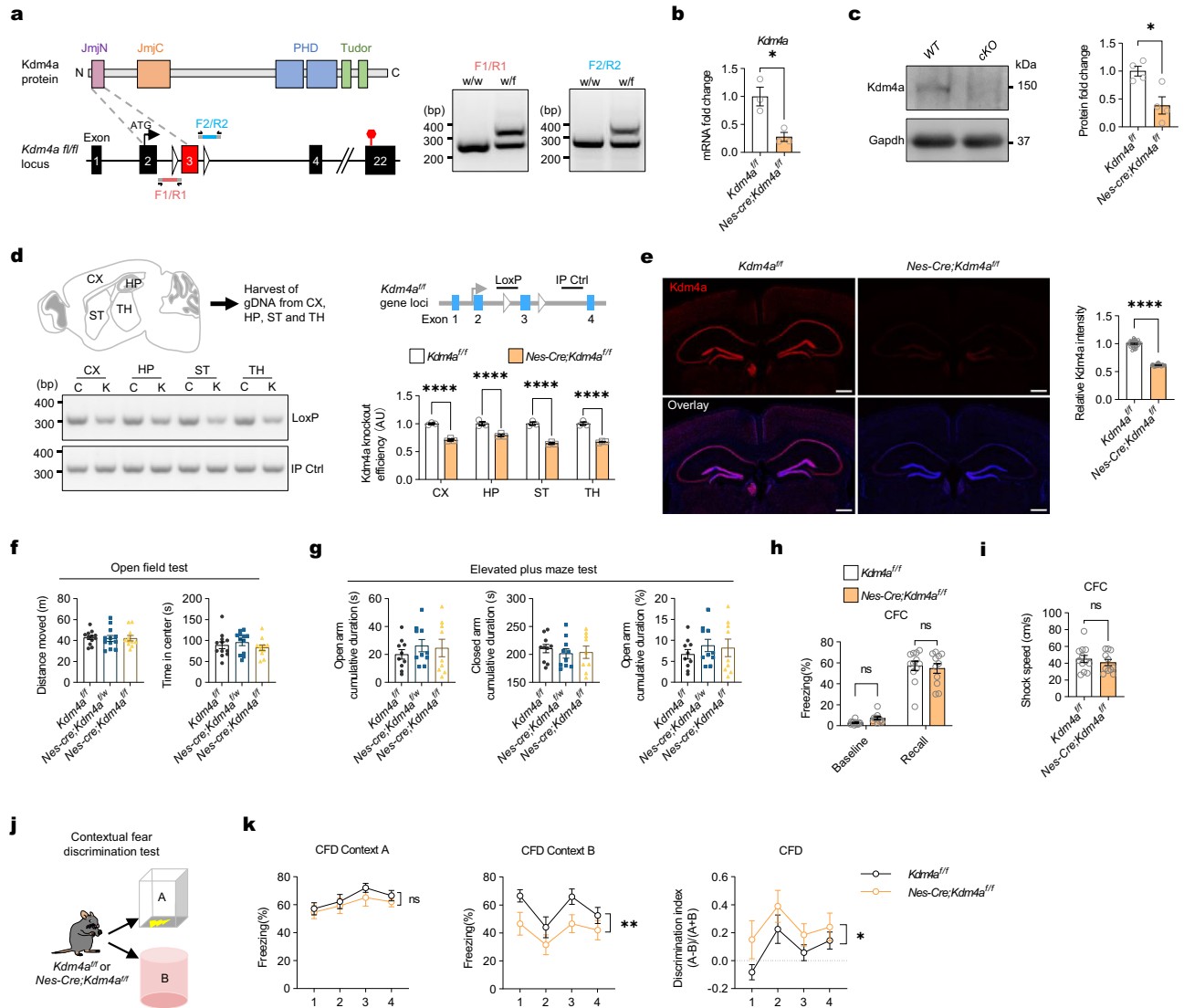

**Fig. 6 | conditional knockout of Kdm4a in DG neurons facilitates memory formation in pattern separation task. a** (Left) Strategy for construction of conditional Kdm4a knockout mice. (Right) PCR identification results of *Kdm4a*^{wt/fl} mice. **b** RT-qPCR analysis of Kdm4a mRNA expression levels in the DG. *Kdm4a*^{f/f}, n = 3; *Nes-Cre;Kdm4a*^{f/f}, n = 3. Two-tailed unpaired t-test, $t_4 = 3.911$, * $p = 0.0174$. Data are presented as mean ± s.e.m. **c** Western blot detection of Kdm4a knockout efficiency in the hippocampus of *Nes-Cre;Kdm4a*^{f/f} mice. *Kdm4a*^{f/f}, n = 4; *Nes-Cre;Kdm4a*^{f/f}, n = 4. Two-tailed unpaired t-test, $t_6 = 3.456$, * $p = 0.0135$. Data are presented as mean ± s.e.m. **d** Genomic DNA PCR detection of Kdm4a knockout efficiency of *Nes-Cre;Kdm4a*^{f/f} mice. CX, cortex; HP, hippocampus; ST, stratum; TH, thalamus. C, control (*Kdm4a*^{f/f}); K, Kdm4a cKO (*Nes-Cre;Kdm4a*^{f/f}). LoxP: Kdm4a loxP primer; IP Ctrl: intrinsic positive control primer. *Kdm4a*^{f/f}, n = 3; *Nes-Cre;Kdm4a*^{f/f}, n = 3. Two-way ANOVA followed by Dunnett test, $F_{1,4} = 113.3$, $p = 0.0004$. **** $p < 0.0001$. Data are presented as mean ± s.e.m. **e** Immunofluorescence staining of Kdm4a in the *Kdm4a*^{f/f} versus *Nes-Cre;Kdm4a*^{f/f} mice. Scale bar, 500 μm. *Kdm4a*^{f/f}, n = 3, region = 31; *Nes-Cre;Kdm4a*^{f/f}, n = 3, region = 28. Two-tailed unpaired t-test, $t_{57} = 34.13$, **** $p < 0.0001$. Data are presented as mean ± s.e.m. **f** Open field test. The locomotor ability of *Kdm4a cKO* mice was normal. *Kdm4a*^{f/f}, n = 12; *Nes-Cre;Kdm4a*^{f/w}, n = 12; *Nes;Cre;Kdm4a*^{f/f}, n = 11. One-way ANOVA followed by Dunnett test, Distance moved: $F_{2,32} = 0.02494$, $p = 0.9754$; Time in center: $F_{2,32} = 0.6745$, $p = 0.5165$. Data are presented as mean ± s.e.m. **g** Elevated plus maze test. Neuron-specific knockout of Kdm4a did not cause

increased anxiety levels in mice. *Kdm4a*^{f/f}, n = 10; *Nes-Cre;Kdm4a*^{f/w}, n = 10; *Nes-Cre;Kdm4a*^{f/f}, n = 9. One-way ANOVA followed by Dunnett test, Time in open arm: $F_{2,26} = 0.4922$, $p = 0.6168$; Time in closed arm: $F_{2,26} = 0.2663$, $p = 0.7682$; Percentage in open arm: $F_{2,26} = 0.492$, $p = 0.6168$. Data are presented as mean ± s.e.m. **h** There was no difference in freezing levels between *Kdm4a cKO* mice and control mice during the memory retrieval phase of contextual fear conditioning. *Kdm4a*^{f/f}, n = 12; *Nes-Cre; Kdm4a*^{f/f}, n = 11. Two-way ANOVA followed by Sidak test, $F_{1,21} = 0.07286$, $p = 0.7898$. Data are presented as mean ± s.e.m. **i,** The shock speed when the mice received footshock was not significantly different between *Kdm4a cKO* mice and control mice. *Kdm4a*^{f/f}, n = 12; *Nes-Cre; Kdm4a*^{f/f}, n = 11. Two-tailed unpaired t-test, $t_{21} = 0.7363$, $p = 0.4697$. Data are presented as mean ± s.e.m. **j** *Kdm4a cKO* mice and control mice were subjected to the contextual fear discrimination test. **k** (Left) There was no difference in freezing levels between *Kdm4a cKO* mice and control mice in context A. *Kdm4a*^{f/f}, n = 12; *Nes-Cre; Kdm4a*^{f/f}, n = 11. Two-way ANOVA followed by Sidak test, $F_{1,21} = 0.9275$, $p = 0.3465$. CFD: contextual fear discrimination. (Middle) The freezing level of *Kdm4a cKO* mice in context B was significantly lower than that of control mice. Two-way ANOVA followed by Sidak test, $F_{1,21} = 11.77$, ** $p = 0.0025$. (Right) The discrimination index of *Kdm4a cKO* mice in the contextual fear discrimination test was significantly higher than that of control mice. Two-way ANOVA followed by Sidak test, $F_{1,21} = 5.497$, * $p = 0.029$. Data are presented as mean ± s.e.m.

## Stereotaxic surgery and viral injection

For delivery of the virus into the DG, mice were anesthetized using isoflurane (4% for induction and 1.5% for maintenance) and mounted in a stereotaxic apparatus. The fur on mouse head was trimmed to expose the skin, which was then sterilized using 70% ethanol followed by 10% iodophor solution. A 1-cm skin incision was performed to find the Bregma point on the skull, and the placement of injection points was determined in relation to Bregma. Burr holes were drilled using a sterile hand drill to expose the brain. After craniotomy, 1 μl of lentivirus or 350 nl of AAV virus was slowly injected (50 nl/min) into the bilateral DG (coordinates from Bregma: −2 mm AP, ±1.5 mm ML, −2 mm DV) using a micro syringe pump. The syringe was left in tissue for 10 min after each injection to allow the viral particles to diffuse. After viral injection, the surgical incision was closed with sutures and the mice were then placed on a heating pad for recovery. Behavioral tasks were conducted at 2 weeks after the surgeries.

In this paper, LV-U6-shCtrl-CMV-EYFP (titer, $2.42 \times 10^9$ TU/ml), LV-U6-shKdm4a-CMV-EYFP (titer, $1.42 \times 10^9$ TU/ml), LV-U6-sgCtrl-EF1α-nls-Cas9-P2A-EGFP (titer, $2.64 \times 10^8$ TU/ml), LV-U6-sgKdm4a-EF1α-nls-Cas9-P2A-EGFP (titer, $2.87 \times 10^8$ TU/ml) were packaged by OBiO Technology (Shanghai) Co., Ltd. AAV2/9-CaMKIIα-hM3Dq-mCherry (titer, $2.67 \times 10^{12}$ genomic copies per ml), AAV2/9-CaMKIIα-mCherry (titer, $2.59 \times 10^{12}$ genomic copies per ml) were purchased from OBiO Technology (Shanghai) Co., Ltd.

## Fluorescence-activated cell sorting (FACS)

Fluorescence-activated cell sorting was performed as previously described[16,28]. Mice were anesthetized using overdosed isoflurane. Mouse brain was perfused with 20 ml ice-cold perfusion buffer (115 mM Choline chloride, 2.5 mM KCl, 1.25 mM $NaH_2PO_4$, 26 mM $NaHCO_3$, 10 mM glucose, 8 mM $MgSO4$, 1 mM Sodium L-ascorbate and 3 mM Sodium pyruvate, pH 7.4) that pre-bubbled with 5% $CO_2$/95% $O_2$ on ice for 30 min. After perfusion, mouse brain was dissected and sectioned using pre-chilled blades and coronal brain matrices for rodents (1 mm thick). Brain sections were immersed in pre-chilled modified EBSS buffer (116 mM NaCl, 5.4 mM KCl, 1 mM $NaH_2PO_4$, 26 mM $NaHCO_3$, 1.5 mM $CaCl_2$, 1 mM $MgSO_4$, 0.5 mM EDTA, 25 mM glucose and 1 mM L-Cysteine, pH 7.4) in a 6-cm cell culture dish. The dentate gyrus was microdissected and rinsed with fresh modified EBSS buffer. Tissue pieces were transferred to a new 15-ml Falcon tube containing 2 ml of modified EBSS buffer and centrifuged at $300 \times g$ for 5 min at 4 °C. After centrifugation, the tissue pieces were gently resuspended with 3 ml of digestion buffer (modified EBSS buffer supplemented with 20 units/ml papain and 0.005% DNase I) and then incubated in a water bath at 37 °C for 45 min with 5% $CO_2$/95% $O_2$. After papain-based enzymatic dissociation, tissues were centrifuged at $300 g$ for 5 min at 4 °C and the pellet was resuspended with 1 ml of stop solution (modified EBSS buffer supplemented with 1 mg/ml Trypsin inhibitor, 1 mg/ml BSA and 0.005% DNase I) to quench papain. The tissues were gently triturated through fire-polished P1000 pipette tips, then fire-polished P200 pipette tips. To remove myelin, cell suspensions were centrifuged at $300 \times g$ for 5 min at 4 °C, resuspended in 1.5 ml of stop solution supplemented with 0.9 M sucrose, and centrifuged at $600 \times g$ for 10 min at 4 °C. The cell pellet was then resuspended with 0.5 ml stop solution and passed through a 100-μm cell strainer to remove debris. 10 μg/ml DAPI was added and incubated on ice in the dark for 5 min before cell sorting. Live cells were sorted using the BD FACS Aria III instrument into 1.5-ml EP tubes containing stop solution. Before sorting, unstained controls and single-color stained controls were used to validate the gating strategies. Dead cells were excluded by staining with DAPI.

For genomic DNA isolation, cells were collected by spinning at $2000 \times g$ for 10 min at 4 °C. The pellets were then snap-frozen and stored at −80 °C until downstream analysis. For RNA extraction, cells were sorted directly into lysis buffer and immediately snap-frozen until reverse transcription.

## Preparation of samples for NGS

For the preparation of sequencing libraries, genomic DNA was extracted using the QIAamp DNA Mini Kit (QIGEN, 51304) with carrier RNA (Poly rA) (QIAGEN, 1068337). The integrated sgRNA sequences were PCR amplified and barcoded using the Q5 Hot Start HiFi DNA Polymerase (NEB, M0515). The libraries were then purified and sequenced using the Illumina HiSeq X system.

## CRISPR screen analysis

For the MAGeCK-Flute[54,55] analysis of the CRISPR screen, two fastq files, named GFP⁻mCherry⁺ and GFP⁺mCherry⁺, were used as input to a Python-based MAGeCK-Flute platform to calculate the read counts and fold changes for GFP⁺mCherry⁺ versus GFP⁻mCherry⁺ to score and rank. The read counts for each sgRNA were normalized as follows:

$$\text{Norm. reads per sgRNA} = \log 2\left(\frac{\text{reads per sgRNA}}{\text{total reads of all sgRNA}} \times \text{num. of sgNRA in library} \times 15 + 1\right)$$

Plasmid fastq file were used as a control for MAGeCK-Flute analysis. P values and gene rankings generated from MAGeCK-Flute analysis were used for identification of candidate genes.

## T7E1 mismatch detection assay

The sgRNAs targeting the mouse *Kdm4a* were designed using the E-CRISP[51] sgRNA designing tool. Kdm4a-targeting sgRNAs were cloned into lentiCRISPRv2-EGFP vectors. Neuro-2a cells were transiently transfected with the plasmids using EZ Trans reagent. EGFP⁺ cells were sorted through flow cytometry. Then, genomic DNAs were extracted and used for PCR amplification of the regions containing sgRNA targeting site. The PCR products were purified and subjected to denaturing and reannealing. The hybridization reaction generates mismatches that are cleaved by the T7 endonuclease 1 (T7E1). Half of the sample was kept as an undigested control. 1 μl of T7 endonuclease 1 was added to the remaining sample and incubated for 15 min at 37 °C. The digested and undigested hybridized DNA samples were resolved by 2% agarose gel electrophoresis.

## TIDE analysis

The genomic DNA was extracted from CRISPR-editing cells, as well as an unedited control. The genomic regions containing sgRNA-targeting site were PCR amplified. The PCR products were purified and used for Sanger sequencing. The sequencing files for the edited and control samples, and 20-nt guide sequence upstream of PAM, were submitted to the TIDE[56] (Tracking of Indels by DEcomposition) or ICE[57] (Inference of CRISPR Edits) web tools for identification of the predominant types of indels and quantification of the editing efficacy.

## Primary neuron culture

Embryonic cortices from *C57BL/6J* mice were dissected at E16.5 and dissociated with 0.25% Trypsin and 0.04% DNase I in the dissection medium (1x HBSS, 10 mM HEPES, 11 mg/ml sodium pyruvate, 0.1% glucose) at 37 °C for 15 min. Digested cortices were centrifuged and gently triturated through fire-polished P1000 pipette with a plating medium (Neurobasal medium, 10% FBS, 1% GlutaMAX Supplement, 1% Pen/Srep). Neurons were plated on cell culture dishes pre-coated with coating solution (20 μg/mL poly-D-lysine, 4 μg/mL laminin) overnight. Neurons were diluted and plated at a density of $10^6$ cells/dish on 3.5-cm dishes. Neurons were grown in maintenance medium (Neurobasal medium, 2% B-27 Supplement, 1% GlutaMAX Supplement, 1%Pen/Srep) at 37 °C with 5% $CO_2$ for 12–14 days. Half of the maintenance medium was changed every 3–4 days.

For virus infection experiments, the lentivirus was added at DIV3 and the infected neurons were collected at DIV10 for RNA isolation and RT-qPCR assay. For high-potassium chloride (KCl) depolarization experiments, DIV14 neurons were incubated with 1/3 volume of depolarization solution (170 mM KCl, 10 mM HEPES, pH7.4, 1 mM $MgCl_2$, 2 mM $CaCl_2$) into the culture medium (final 55 mM KCl) for 1 h to achieve neuronal activation. Neurons were collected after 0 h or 1 h KCl-induced depolarization.

### Imaging and biochemical assays

**Fluorescent immunostaining.** Cultured cells were fixed using 4% paraformaldehyde (PFA) in PBS at 4 °C for 10 min. Mice were anesthetized using an overdose of isoflurane. The mouse brain was perfused and fixed with 4% PFA in PBS at 4 °C overnight. Samples were sectioned coronally (40 μm-thickness) on a Leica Vibrating blade microtome. Fixed cells and brain slices were blocked and permeabilized with blocking buffer (1% bovine serum albumin, 0.3% Triton X-100 in PBS, pH 7.4) at 4 °C for 1 h, followed by incubation with primary antibodies diluted in the blocking buffer at 4 °C overnight with gentle shaking. After washing, secondary antibodies diluted in the blocking buffer were applied to the samples at room temperature for 2 h. Nuclei were stained by incubation with 0.5 μg/mL 4,6-diamidine-2-phenylindole (DAPI) in PBS at room temperature for 10 min. The samples were washed thrice with PBS for 15 min each time. The stained samples were mounted with anti-fade mounting medium and used for confocal microscopy.

**Imaging analysis.** Images were taken with a spinning disk confocal microscope (Nikon, CSU-W1 SoRa). Images were acquired with 20× or 60× objectives at identical acquisition settings. The number of IEG-expressing cells, the intensity of fluorescent signals, the density and size of Kdm4a puncta, the density and morphology of spine and axonal boutons were quantified using ImageJ (1.53q) and Icy (v2.4.3.0).

**Quantification of IEG-expressing cells.** The whole DG imaging was obtained with a 60× oil immersion object. The Egr1+, Fos+, EYFP+ cells and double-positive cells in the granule cell layer (GCL) were counted (co-expression of Egr1+EYFP+ or Fos+EYFP+). Then divide it by the area of GCL.

**Quantification of nuclear Kdm4a signals.** The images were acquired using a 60x oil immersion object and SoRa super-resolution module. The cultured neurons were immunostained with NeuN and Kdm4a. The Kdm4a signals were measured in the NeuN+ nuclei. The Kdm4a fluorescence intensity per nucleus were measured and divided by the area. The Kdm4a punctas per nucleus were counted and the intensity of Kdm4a punctas were quantified.

**Dendritic spine morphology.** The mashroom spines of DG GCs in the molecular layer were identified using the spine head width (H) and spine nech width (N). Mashroom spines: H > 2 N. The spine density per 10 μm was quantified.

**Axonal bouton morphology.** The mossy fiber (MF) boutons of DG GCs in the stratum lucidum (sl) of CA3 were identified using the EYFP signals. The area of MF boutons were quantified.

**RNA extraction and RT-qPCR.** Cultured cells or mouse brain tissues that were dissected were homogenized using 0.2 ml of TRIzol reagent (Invitrogen) and incubated at room temperature for 5 min. Add 40 μl chloroform to the cell lysate and vortex the samples vigorously. RNA remained in the aqueous phase after centrifugation at $12,000 \times g$ for 15 min at 4 °C. The upper aqueous phase that contained RNA was collected for precipitation by adding 0.1 ml of isopropyl alcohol.

Samples were mixed and incubated at room temperature for 10 min and centrifuged at $12,000 \times g$ for 10 min at 4 °C. The RNA pellet was washed twice with 0.1 ml of 75% ethanol and centrifuged at $12,000 \times g$ for 5 min at 4 °C. The RNA pellet was air-dried for 5 min and dissolved in 20 μl of DEPC-treated $H_2O$. The concentration of RNA was determined by a spectrophotometer at 260 and 280 nm.

Reverse transcription was performed using the HiScript III 1st Strand cDNA Synthesis Kit (Vazyme, R312-01). The diluted cDNAs (at a ratio of 1:10) were used for SYBR Geen (Bimake, B21202)-based quantitative real-time PCR to determine the relative expression levels of genes of interest using the ΔΔCt quantification method. All RT-qCPR primers used in this study were outlined in Supplementary Data 6.

**RNA-seq analysis.** For the preparation of sequencing samples, lentivirus LV-U6-shKdm4a-CMV-EYFP or LV-U6-shCtrl-CMV-EYFP were delivered into the adult male *C57BL/6J* mouse DG through stereotaxic injection. After 14 days of recovery, mice were anesthetized using overdosed isoflurane and perfused with ice-cold PBS. The DG was microdissected and dissociated with papain-based enzymatic digestion. 2,000 EYFP+ cells were sorted using the BD FACS Aria III instrument into a 200-μl PCR tube containing lysis buffer and RNase inhibitor. After sorting, the tubes were immediately snap-frozen. Total RNA was reverse transcribed using the SMART-Seq HT Kit (Clontech, 634437), and cDNAs were used to produce the sequencing library. The libraries were purified and sequenced using the Illumina NovaSeq 6000 system.

For RNA-seq data analysis, raw reads were filtered using Seqtk (v.1.4) to remove adapters and low-quality regions, and mapped to the GRCm38 genome using Hista2[58] (v.2.0.4). Reads on each gene were counted by Stringtie[59] (v.1.3.0). Differential expressed genes (DEGs) were identified using edgeR. Gene ontology analysis was performed by online GESA[60] tools using all significant DEGs ($P < 0.05$).

**Western blot.** The proteins in cell lysates were resolved by 6%, 10%, or 12% SDS-polyacrylamide gel electrophoresis, depending on the molecular weight of the proteins of interest, and transferred to a polyvinylidene difluoride membrane. The membrane was blocked with 5% skimmed milk in TBST buffer (20 mM Tris, 150 mM NaCl, 0.1% Tween-20, pH 7.6) for 1 h. The membrane was washed three times with TBST buffer and incubated with primary antibodies diluted in the TBST buffer with 5% bovine serum albumin (BSA) overnight with gentle shaking at 4 °C. After washing, HRP-conjugated secondary antibodies diluted in TBST buffer with 5% BSA were applied to the membrane at room temperature for 1 h. The membrane was then washed three times with TBST buffer for 15 min. The signals on the membrane were detected in the ChemiDoc imaging system (Bio-Rad). All antibodies used in western blots were outlined in Supplementary Data 7. The uncropped scans of all blots in Figures were supplied in Supplementary Figs. 15, 16.

**Chromatin immunoprecipitation (ChIP).** For each sample, 20 μl of Protein-G magnetic beads were washed twice with beads binding buffer (1× PBS, pH 7.4, 0.2% Tween-20). The washed beads were incubated with 2 μg of antibodies in 200 μl of beads binding buffer overnight at 4 °C on a rotator. Bead-antibody complexes were washed thrice with beads binding buffer and then washed twice with dilution buffer (0.01% SDS, 1.1% Triton X-100, 1.2 mM EDTA, pH 8.0, 16.7 mM Tris, pH 8.1, 167 mM NaCl). Washed bead-antibody complexes were added with 50 μl dilution buffer and stored at 4 °C until cells were ready.

Dissected mouse brain tissues were dissociated using a homogenizer with 1 ml of ice-cold PBS supplemented with 1x Protease inhibitors cocktail. The lysates were crosslinked with 27 μl of 37% paraformaldehyde (final 1% PFA) for 15 min at room temperature on a rotator. 83.3 μl of 2 M glycine was added (final 150 mM glycine) and

incubated on a rotator for 10 min to quench the crosslink reaction. Cells were collected by centrifugation at 2000 × $g$ for 10 min at 4 °C. The cell pellets were washed and resuspended with 300 µl cell lysis buffer (50 mM Tris, pH 8.0, 140 mM NaCl, 1 mM EDTA, pH 8.0, 10% glycerol, 0.5% NP-40, 0.25% Triton X-100 supplemented with 1× Protease inhibitors cocktail) for a 20 min of incubation on ice. Nuclei were collected by centrifugation at 2000 × $g$ for 5 min at 4 °C. The nuclear pellets were resuspended with 300 µl nuclear lysis buffer (10 mM Tris, pH 8.0, 1 mM EDTA, pH 8.0, 0.5 mM EGTA, pH 8.0, 0.5% SDS supplemented with 1× Protease inhibitors cocktail). After incubation for 20 min on ice, lysates were sonicated at 9 cycles of 50% power 30 s on/off treatment at 4 °C to fragment the chromatin. The samples were centrifuged at 16,000 × $g$ for 10 min at 4 °C to remove insoluble debris. The supernatants were collected and stored on ice. Part of the supernatants were kept as input and stored at −80 °C. Next, the remaining supernatants were added and incubated with 50 µl bead-antibody complexes overnight on a rotator at 4 °C. Then, the beads were washed five times with wash buffer (100 mM Tris, pH 8.0, 500 mM LiCl, 1% NP-40, 1% Sodium deoxycholate) and washed once with TE buffer (50 mM Tris, pH 8.0, 10 mM EDTA). The washed beads were then resuspended in 170 µl elution buffer (50 mM Tris, pH 8.0, 10 mM EDTA, 1% SDS) and incubated for 20 min at 65 °C. The eluted samples and input samples were both incubated in the elution buffer overnight at 65 °C for decrosslinking. Then, 1 µl 10 mg/ml RNase A was added in each sample and incubated at 37 °C for 1 h to remove RNA. 1 µl 20 mg/ml Proteinase K was added and incubated at 55 °C for 2 h to digest protein. Next, the input and ChIP DNAs were extracted using phenol/chloroform/isoamyl alcohol (25:24:1). The purified DNAs were measured on a Nanodrop and then used for real-time qPCR analysis.

**Proximity labeling assay.** In proximity labeling assay, the HEK293T cells transfected with KDM4A-PafA and 3x NLS-Bio-PupE (KDM4A-PUPIT) plasmids were treated with 4 µM biotin for a 24 h of incubation. Control cells were transfected with 3xNLS-PafA and 3x NLS-Bio-PupE (CTRL-PUPIT) plasmids. The nuclear-pupylated proteins were extracted with RIPA lysis buffer (50 mM Tris, pH 7.4, 150 mM NaCl, 1% Triton X-100, 1% Sodium deoxycholate, 0.1% SDS, 5 mM EDTA supplemented with 1x Protease inhibitors cocktail and 1x Phosphatase inhibitors cocktail) and then used for streptavidin immunoprecipitation. The enriched proteins were incubated at 95 °C in 2x Laemmli loading buffer for 10 min and then subjected to western blot or mass spectrometry analysis.

**Mass spectrometry analysis.** For the preparation of MS samples, the enriched proteins were first resolved by 10% SDS-PAGE, and electrophoresing for 2 cm in the gel. After electrophoresis, the gel was washed thrice with ddH$_2$O for 15 min. For each sample, the entire region containing proteins were excised and cut into 1 mm slices, and then collected in a 1.5-ml tube. After washing with ddH$_2$O, the gel slices were incubated with 0.5 ml acetonitrile (ACN) to dry up the gel pieces at 37 °C for 10 min at 500 rpm, and repeat this step once. Completely dehydrate the gel slices using a vacuum centrifuge for 10 min. 0.5 ml of 10 mM DTT in 25 mM NH$_4$HCO$_3$ solution were added in the gel slices and incubated at 65 °C for 1 h at 500 rpm. The supernatants were removed, and 0.4 ml of 50 mM iodoacetamide (IAM) in 25 mM NH$_4$HCO$_3$ solution was then added for alkylation, and incubated in the dark at room temperature for 45 min. The supernatants were replaced by 25 mM NH$_4$HCO$_3$ solution, and incubated at room temperature for 10 min on a rotator. After the removal of supernatants, the same volume of 25 mM NH$_4$HCO$_3$ solution and ACN were added in the gel slices and incubated at 37 °C for 5 min at 500 rpm. The supernatants were replaced by 0.5 ml of ACN, and incubated at 37 °C for 10 min at 500 rpm. Completely dehydrate the gel slices using a vacuum centrifuge for 10 min. For each sample, 3 µl of Trypsin (Promega, v5113) and 0.5 ml 25 mM NH$_4$HCO$_3$ solution were added in gel slices and then

digested at 37 °C overnight at 500 rpm. On the following day, the digested proteins were centrifuged at 3000 × $g$ at room temperature for 3 min. For each sample, 400 µl of 50% ACN, 0.5% formic acid (FA) solution was added and incubated at 37 °C for 10 min at 500 rpm. The supernatants were transferred into a new 1.5-ml tube, and repeat this step once with 200 µl of 50% ACN, 0.5% FA solution. Completely dehydrate the combined supernatants using a vacuum centrifuge for 3–4 h. After dehydration, 50 µl of 0.1% FA solution was added to dissolve the peptides. To remove the salt from sample, Pierce C18 tips (Thermo, SP301) were washed twice with 200 µl buffer-B (50% ACN, 0.1% FA), and centrifuged at 500 × $g$ for 3 min. The tips were then washed twice with 100 µl buffer-A (0.1% FA), and centrifuged at 500 × $g$ for 3 min. The washed tip was transferred to a new 1.5-ml tube. Then, the solution containing peptides was injected into the Pierce C18 tips and centrifuged at 500 × $g$ for 3 min, and then collected the follow-through. The follow-through was re-injected into the tips and centrifuged again. The tips containing peptides were washed with 100 µl buffer-A and then transferred to a new 1.5-ml tube. for elution of peptides, 50 µl of buffer-B was added into the tips to elute the peptides that bind to the membrane in the tip, and centrifuged at 500 × $g$ for 3 min. This step was repeated once. The eluted peptides were completely dehydrated using a vacuum centrifuge for 1 h. The sample was then dissolved in 10 µl of ACN solution, and 3 µl of the solution was injected into the LC-MS/MS instruments. The peptides were identified and quantified using MaxQuant[61] and Perseus[62] platforms.

**Co-immunoprecipitation (Co-IP).** About 10[7] HEK293T cells that were stably co-transfected with Flag-KDM4A and HA-YTHDC2 plasmids were used to detect the interaction between KDM4A and YTHDC2. Cells were incubated with 4 ml crosslink solution (1 mM DSP in PBS, pH 7.2) in 10-cm cell cultured dishes at room temperature for 30 min. The same volume of stop solution (40 mM Tris, pH 7.5 in PBS) was added (final 20 mM Tris) and incubated at room temperature for 15 min to quench the crosslink reaction. Cells were collected and lysed with Co-IP lysis buffer (50 mM Tris, pH 7.4, 150 mM NaCl, 1% Triton X-100, 1% Sodium deoxycholate, 0.1% SDS, 5 mM EDTA and 1x Protease inhibitor cocktail) for 20 min on ice. The lysate was centrifuged to remove cell debris at 15,000 × $g$ for 10 min at 4 °C. The supernatants were then collected and kept on ice. Protein concentration was measured using BCA Kit (Beyotime, P0010S). Part of the protein samples were kept as input and stored at −80 °C.

For each protein immunoprecipitation, 30 µl pre-washed protein G magnetic beads incubated with 3 µg of antibodies (anti-Flag, anti-HA and anti-IgG respectively) overnight on a rotator at 4 °C. Then, the washed antibody-beads complexes were added to each remaining supernatant, followed by incubation overnight at 4 °C on a rotator. The beads were collected and washed five times with Co-IP lysis buffer. The enriched proteins were eluted with 3× Laemmli loading buffer. The input and IP samples were incubated at 95 °C in Laemmli loading buffer for 10 min and then used for western blot analysis.

**Nuclear run-on (NRO) assay.** For each replicate, 5 × 10[6] Neuro-2a cells were collected and washed twice with ice-cold PBS. Cells were collected by centrifugation and the pellet was resuspended in 1 ml of cell lysis buffer (10 mM Tris-HCl, pH 7.4, 10 mM NaCl, 3 mM MgCl$_2$, 0.5% NP-40 supplemented with 40 U/ml RNase inhibitor and 1× Protease inhibitors cocktail). After a 5 min incubation on ice, the lysates were centrifuged at 300 × $g$ for 5 min at 4 °C to collect nuclei. The nuclear pellet was then resuspended in 0.5 ml nuclear lysis buffer (10 mM Tris-HCl, pH 7.4, 3 mM CaCl$_2$, 2 mM MgCl$_2$, 0.5% NP-40, 10% glycerol supplemented with 40 U/ml RNase inhibitor and 1x Protease inhibitors cocktail). The nuclear lysates were then centrifuged and the pellet was resuspended in 40 µl nuclear storage buffer (50 mM Tris-HCl, pH 8.3, 0.1 mM EDTA, 5 mM MgCl$_2$ and 40% glycerol) and kept on ice. For each

sample, 60 μl of run-on buffer (10 mM Tris-HCl, pH 8.0, 1 mM DTT, 5 mM MgCl₂, 0.3 M KCl, 1% Sarkosyl, 100 U/ml RNase inhibitor supplemented with 1 mM ATP/GTP/CTP, 0.5 mM UTP and Br-UTP) was added in the lysates and mixed gently. The reaction mix was incubated in a water-bath at 30 °C for 30 min. After run-on reaction, 12 μl of 2× DNase I was added for another 5 min of incubation at 30 °C to remove DNA. Protein digestion buffer (20 mM Tris-HCl, pH 7.4, 2% SDS, 10 mM EDTA supplemented with 200 μg/ml Proteinase K) of the same volume was added in the run-on sample and incubated at 55 °C for 1 h to digest protein. RNAs were purified from samples using phenol/chloroform extraction and eluted in 100 μl of DEPC-treated H₂O. For nascent RNA immunoprecipitation, 30 μl of Protein-G magnetic beads were washed thrice with PBST buffer (0.1% Tween-20 in PBS) and resuspended in 30 μl PBST buffer. Beads were incubated with 2 μg of Mouse anti-BrdU antibody (Sigma, B2531-.2 ML) at room temperature for 10 min. Blocking buffer (0.1% PVP, 1 mg/ml BSA in PBST buffer) was added in the bead-antibody complexes and incubated for another 30 min of incubation. Next, bead-antibody complexes were washed twice with PBSTR (PBST buffer supplemented with 40 U/ml RNase inhibitor) and resuspended in 100 μl PBSTR. RNAs extracted from run-on reaction were incubated at 65 °C for 5 min to open their secondary structure, and then added in the bead-antibody complexes. After 30 min of incubation, beads were collected and washed thrice with PBSTR. Then, the enriched nascent RNAs were eluted using 0.5 ml of TRIzol reagent and isolated by phenol/chloroform extraction. 10 μl of the purified nascent RNAs were used for reverse transcription and RT-qPCR analysis.

**RNA Immunoprecipitation (RIP).** Dissected mouse brain tissues were dissociated using a homogenizer in 1 ml RIP buffer (50 mM Tris, pH 7.5, 150 mM NaCl, 1 mM EDTA, pH 8.0, 1 mM DTT, 0.5% NP-40 supplemented with 40 U/ml RNase inhibitor and 1x Protease inhibitors cocktail) for 20 min on ice with gentle shanking. The lysates were centrifuged to remove cell debris at 16,000 × g at 4 °C for 10 min. The supernatants were then collected and kept on ice. Part of the supernatants were kept as input and stored at −80 °C. The remaining supernatants were incubated with 4 μg of anti-Ythdc2 antibody or control IgG at 4 °C overnight on a rotator. Pre-washed Protein-G magnetic beads were added to each antibody-lysate complex, followed by 3 h of incubation. Then, the beads-antibody-RNA complexes were collected and washed three times with RIP lysis buffer at 4 °C. The enriched RIP-RNAs were eluted using 0.5 ml TRIzol reagent for RNA isolation. RIP-RNA and Input RNA samples were both incubated with TRIzol reagent on ice for 5 min. Add 100 μl chloroform in RNA samples and vortex samples vigorously, followed by 5 min of incubation. RNA remained in the aqueous phase after centrifugation at 15,000 × g for 10 min at 4 °C. The upper aqueous phase that contained RNA was collected for precipitation by adding 20 μl 3 M Sodium acetate, pH 5.2, 1 μl 20 mg/ml Glycogen and 500 μl 100% ethanol. Samples were mixed and incubated overnight at −80 °C. Samples were then centrifuged at 15,000 × g for 25 min at 4 °C. The RNA pellet was washed with 0.2 ml 75% ethanol and centrifuged at 15,000 × g for 15 min at 4 °C. The RNA pellet was air-dried for 5 min and dissolved in 20 μl DEPC-treated H₂O. 10 μl of each RNA sample were used for reverse transcription and RT-qPCR analysis.

**m⁶A immunoprecipitation.** For each sample, 20 μg of total RNA that was harvested from the mouse brain was incubated with RNA fragmentation buffer (10 mM Tris-HCl, pH 7.5, 10 mM ZnCl₂ in DEPC-treated H₂O) in a 200-μl PCR tube at 94 °C for 4 min to fragment RNAs into about 200-nt-long fragments. After incubation, the samples were transferred on ice immediately. 2 μl of 0.5 M EDTA solution was added to the samples to stop the fragmentation. The purified fragmented RNAs were eluted in 100 μl DEPC-treated H₂O. Part of the RNA samples were kept as input and stored at −80 °C. The remaining fragmented RNAs were diluted with 1x IP buffer (10 mM Tris-HCl, pH 7.5, 150 mM NaCl, 0.1% NP-40 in DEPC-treated H₂O supplemented with 40 U/ml

RNase inhibitor and 2 mM RVC). 5 μl of rabbit anti-m6A antibody or control IgG were added to the RNA samples, which were then incubated for 2 h at 4 °C on a rotator. 30 μl of Protein-G magnetic beads were washed twice with 1x IP buffer and then blocked with 0.5 mg/ml BSA in 1× IP buffer for 2 h at 4 °C. The beads were then washed twice with 1× IP buffer and added to the antibody-RNA complexes for another 4 h of incubation at 4 °C. The beads-antibody-RNA complexes were collected and washed three times with 1x IP buffer at 4 °C. The enriched m⁶A-RIP-RNAs were eluted using 0.5 ml TRIzol reagent for RNA isolation. The purified m⁶A RNA and input RNA samples were dissolved in 20 μl DEPC-treated H₂O. 10 μl of each RNA sample were used for reverse transcription and RT-qPCR analysis.

**Dual-luciferase reporter assay.** To generate Trpm7 WT renilla luciferase reporter vector, the fragment of exon 12 (from mouse *Trpm7* CDS) containing the predicted m⁶A sites was PCR amplified from Nuro-2a cell cDNA library and cloned into the psiCheck2 vector between the restriction enzyme sites Not I and Xho I. The putative two m⁶A sites on *Trpm7* exon 12 were then replaced by Thymine (T) using site-directed overlapping PCR mutagenesis to generate Trpm7 Mut reporter vector. Full-length Ythdc2 CDS fragments were amplified by PCR and cloned into pcDNA3.1 vector using Gibson Assembly.

The Neuro-2a cells were seeded in a 96-well plate and maintained until they reached 80% confluency. 100 ng of luciferase reporter vectors (Trpm7 WT or Mut) and 100 ng of Ythdc2 CDS or mock vectors were co-transfected using EZ Trans Reagent (Life-iLab). After 36 hrs of transfection, the cells were lysed with 100 μl of lysis buffer (Beyotime) for 15 min at room temperature with gentle shanking. The lysates were centrifugated at 12,000 × g for 5 min to remove cell debris. 80 μl of the supernatants were used for measurement of firefly (Fluc) and renilla (Rluc) luciferase activities with Dual Luciferase Reporter Gene Assay Kit II (Beyotime, RG029S) in the SpectraMax i3 microplate reader (1 s per well). The relative luciferase activity was calculated by dividing Fluc by Rluc and normalized to control. For measuring the Rluc and Fluc mRNA levels, the cells were lysed with TRIzol reagent and total RNAs were purified for RT-qPCR analysis. The relative luciferase mRNA changes were calculated as described above.

**Split-luciferase complementation assay.** To generate the Kdm4a-Nluc/Nluc-Kdm4a and Ythdc2-Cluc/Cluc-Ythdc2 expression plasmids, the N-terminal domain of firefly luciferase (amino acids 2-416) was cloned into either N- or C-terminal of pcDNA-Flag-Kdm4a plasmid (from this paper), and the C-terminal domain of firefly luciferase (amino acids 389-550) was cloned into either N- or C-terminal of pcDNA-Ythdc2-Myc plasmid (from this paper) through Gibson Assembly.

Neuo-2a cells were seeded in white 96-well plates and maintained until they reached 80% confluency. Plasmids harboring the split-luciferase fusions (e.g., Nluc-Kdm4a and Cluc-Ythdc2) were co-transfected in 1:1 ratio (200 ng of plasmids total per well) using EZ Trans Reagent (Life-iLab) according to the manufacturer's instructions. After 36 hrs of transfection, the culture medium was removed and washed with prewarmed PBS. The medium was replaced with 100 μl of prewarmed Fluc imaging buffer (1x PBS, 1 mM MgCl₂, 0.9 mM CaCl₂, 0.1% Glucose, 150 mg/ml D-luciferin). The cells were then imaged in the SpectraMax i3 microplate reader (1 s per well).

**Fluorescence-activated nuclei sorting (FANS).** FANS was performed as previously described with some modifications[2,6]. Mice were anesthetized using overdosed isoflurane. Mouse brain was dissected and immersed in the pre-chilled modified EBSS buffer. The dentate gyrus was excised and placed into an pre-chilled 2-ml glass Dounce grinder tube (Sigma-Aldrich T2690-1EA) with 0.5 ml of ice-cold Nuclei isolation buffer (250 mM Sucrose, 25 mM KCl, 5 mM MgCl₂, 10 mM Tris-HCl (pH 7.4), 0.1% Triton X-100, 1 mM DTT supplemented with 30 U/ml RNase inhibitor and 1x Protease inhibitor cocktails). The tissue was gently

homogenized on ice with the pestle (Sigma-Aldrich P0485-1EA) 30 times. The homogenate was transferred to a pre-chilled 1.5-ml EP tube and incubated on ice for 7 min, then centrifuged at 500 g for 5 min at 4 °C to pellet the nuclei. The pellet was resuspended in 1 ml of ice-cold Nuclei storage buffer (250 mM Sucrose, 5 mM MgCl$_2$, 10 mM Tris-HCl (pH 7.4), 1 mM DTT, 1% BSA supplemented with 30 U/ml RNase inhibitor and 1x Protease inhibitor cocktails), and centrifuged at 500 g for 5 min at 4 °C. The pellet was then resuspended in 0.5 ml of ice-cold Nuclei storage buffer and filtered through 40-μm cell strainer. The nuclei suspension was incubated on ice for 30 min, then incubated with Mouse anti-Fos (1:1000) and Rabbit anti-NeuN (1:400) primary antibodies on a raotator for 45 min at 4 °C. The nuclei were centrifuged at 500 g for 5 min at 4 °C and then washed with Nuclei storage buffer. The washed nuclei suspension was incubated with 5 μM Hoechst-33342 and anti-Mouse (1:500, conjugated with Alexa Fluor 555) and anti-Rabbit (1:500, conjugated with Alexa Fluor 488) secondary antibodies on a rotator for 30 min at 4 °C. The stained nuclei were centrifuged at 500 g for 5 min at 4 °C and washed with Nuclei storage buffer, and then resuspened with 0.5 ml of ice-cold Nuclei storage buffer. NeuN+Fos- and NeuN+Fos+ cells were separately isolated using a BD FACS Aria III cell sorter, and sorted in 0.2-ml PCR tubes preloaded with 3 μl of lysis buffer.

**Reverse transcription of low-input RNA from FACS-sorted cells.** Reverse transcription (RT) of low-input RNA was performed as previously described with some modifications[63,64]. About 350-500 cells/nuclei were sorted directly into 3 μl of Cell lysis buffer (0.2% Triton X-100, 2 U/μl RNase inhibitor in nuclease-free H$_2$O) using a FACS flow cytometer. After sorting, the samples were spin down and transferred on ice immediately. The lysate was incubated with 1 μl of 10 μM Oligo-dT$_{30}$VN primers and 1 μl of 10 μM dNTPs at 72 °C for 3 min, and immediately placed back on ice. RT mixtures (2 μl of 5x Superscript II first-strand buffer, 2 μl of 5 M Betaine, 0.5 μl of 100 mM DTT, 0.25 μl of 40 U/μl RNase inhibitor, 0.1 μl of 100 μM template-switching oligos (TSO) primer, 0.06 μl of 1 M MgCl$_2$ and 0.5 μl of 200 U/μl SuperScript II reverse transcriptase) were added to the samples to obtain a final reaction volume of 12 μl. The mixtures were gently pipetted up and down and spin down, then incubated in a thermal cycler to obtain the first-strand cDNA reaction (42 °C 90 min, (50 °C 2 min, 42 °C 2 min) x 10 cycles, 70 °C 10 min, 4 °C hold). The PCR products were then incubated with 12.5 μl of 2x KAPA HiFi HotStart ReadyMix and 0.5 μl of 10 μM ISPCR primers in a thermal cycler to obtain the full-length cDNA (98 °C 3 min, (98 °C 20 s, 67 °C 15 s, 72 °C 6 min) × 18 cycles, 72 °C 10 min, 4 °C hold). The diluted cDNA products (at a ratio of 1:5) can be directly used for real-time qPCR analysis.

**Generation of Kdm4a$^{fl/fl}$ mice.** Kdm4a$^{fl/fl}$ transgenic mice were generated by using the CRISPR/Cas9 system. Exon 3 of mouse Kdm4a was targeted by inserting a loxP site in intron 2–3 and a loxP site in intron 3-4. A DNA donor containing targeted exon flanked by two loxP sites and the CRISPR/Cas9 systems were microinjected into the C57BL/6J mouse fertilized eggs. Edited eggs were transplanted to obtain positive offspring, which were determined using PCR amplification and Sanger sequencing. Chimeric mice were then bred to wild-type C57BL/6J mouse to establish a stable transgenic mouse line carrying the loxP sites.

### Behavioral tests
**Open field.** Mice were placed in the center of a 46 × 46 cm box and allowed to freely explore for 10 min after 3 min of habituation. Videos were recorded and analyzed by TSE apparatus and software. The 20 × 20 cm region of the box was defined as central zone. The time in center and the total distance traveled were analyzed for evaluating the mouse locomotor activity.

**Contextual fear conditioning.** Mice were placed in the footshock chamber (20 × 20 cm) with stainless steel grid floor, and habituated for 3 min. After habituation, mice received a 1 s foot shock of 0.6 mA,

followed by 30 s of delay. The next day (24 h later), mice were returned to the footshock chamber without shocks for 3 min of exploration (recall). Mouse activity was analyzed by TSE apparatus and software. The percentage of freezing during recall phase was analyzed to determine the ability of fear memory formation in mouse.

**Fear memory extinction.** Mice were subjected to contextual fear conditioning and then returned to the home cage after the recall session. Two weeks later, mice were re-expose to the same footshock chamber without receiving foot shock for 3 min. After exploration, mice were returned to their home cage. This procedure was repeated for 6 days. The memory extinction curve was analyzed for access the ability of active forgetting in mouse.

**Contextual fear discrimination.** To test the ability of pattern separation, mice were subjected to a contextual fear discrimination task. Briefly, mice were placed daily in two similar contexts: context A and context B. The footshock chamber (context A) and no-shock chamber (context B) shared the same stainless steel grid flooring. Context A was a square box and illuminated with white light. Context B was a circular box and illuminated with yellow light, and 0.5% acetic acid odor were delivered below the floor. 75% ethanol were used to clean grid floor before the mouse was placed in the context. The mice were first placed in context A and received a 1 s foot shock of 0.4 mA after 3 min of exploration on day 0. From day 1 to day 4, mice were randomly placed in either context A or context B for 3 min, and then returned to their home cage. 2 h after the first exploration, mice were placed in the other context. The freezing levels in context A and context B each day were calculated and used to determine the discrimination index: (Freezing$_A$ − Freezing$_B$)/(Freezing$_A$ + Freezing$_B$).

**Elevated plus maze.** The elevated plus maze consists of two open arms and two closed arms (30 cm long and 6.5 cm wide), both of which are 55 cm above the floor. Mice were placed in the central area and allowed to freely explore the maze for 5 min. Mouse activity was recorded by a camera above the maze and analyzed using a mouse tracking system (Noldus). The time that mice spent in the open arms and closed arms, and a percentage of being in open arms, were calculated to assess anxiety-like behavior.

**Memory allocation test.** Before the test, mice were habituated to transportation and experimental room for three days. Then, mice were subjected to freely explore three distinct, novel contexts (20 × 20 cm) which were separated by 7 days or 5 h. During the exploration session (3 min), footshock was not given to the mice. Contexts were different in the shape, wall, flooring, lighting and odor. Context A: stainless steel grid floor, black plastic wall, square shape, 75% ethanol, 20% white light; Context B: black plastic floor, transparent plastic wall with vertical stripes, cicular shape, 0.2% acetic acid, 30% yellow light; Context C: white plastic flooring, transparent plastic wall with spots, square shape, 50% yellow light.

Two days later, mice were placed in context A to habituate for 10 s, and then given an immediate footshock (0.5 mA, 2 s). Thirty seconds after the shock, mice were placed back to the homecage. Two days after the footshock, mice were tested in either context A (shocked context), B (5 h; not shocked), or C (7 days; not shocked) for 3 min. The freezing levels were recorded by the TSE apparatus.

**Immediate shock test.** Mice were subjected to two distinct, novel contexts (context A and context B, described above). Mice were placed in context A for 10 s, and then given an immediate footshock (0.5 mA, 2 s). After 30 s of delay, mice were placed back to the homecage. 5 h later mice were placed in context B for another 3 min of exploration. The freezing levels in the delay session of context A and context B were quantified.

**Behavioral analysis.** All animal behaviors in the contexts were recorded and analyzed using the computerized TSE Multi Conditioning System 2.0 (TSE 2.0). The location of mouse was detected by the horizontal and depth infrared photobeams. Analysis parameters used in the software were defined as follows. Grooming >3 s, freezing >1000 ms, activity >3.0 cm/s.

## Statistics

Statistic analysis was performed using GraphPad Prism (v.9.0.0) software. Data were represented as mean ± s.e.m. Box or violin plots were represented as the mean, interquartile range and minimum and maximum. Statistical significance was determined by unpaired or paired two-tailed Student's $t$ test, one or two-way ANOVA. *, $P < 0.05$; **, $P < 0.01$; ***, $P < 0.001$; ****, $P < 0.0001$.

## Reporting summary

Further information on research design is available in the Nature Portfolio Reporting Summary linked to this article.

## Data availability

The RNA-seq data generated in this study have been deposited in the NCBI Gene Expression Omnibus (GEO) database under accession code GSE269325: In vivo neuronal activity downregulated gene lists were downloaded from GEO database under accession code GSE77067 (novelty exploration, ref. 2), GSE82013 (electroconvulsive stimulation, ref. 19) and GSE125068 (kainic acid, ref. 33). CRISPR library, CRISPR screen results, activity-downregulated gene lists, processed RNA-seq data, LC-MS analysis data, primers and antibodies used in this study are provided in the Supplementary Data (Tables 1–7). The Supplementary Data are available on figshare: https://doi.org/10.6084/m9.figshare.25990495. Source data were provided in this paper. Source data are provided in this paper.

## Code availability

Quality control of the plasmid sgRNA library was performed using custom MATLAB codes. All the other sequencing data were analyzed using published protocols and codes described in the Methods sections. The custom codes are available on Zendo: https://zenodo.org/records/11523126.

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

## Acknowledgements

We thank Dr. Yong Cang, Min Long for gifting and helping with CRSIPR library; Dr. Min Zhuang for sharing the PUP-IT proximity labeling plasmids; Dr. Guangyu Wang for MATLAB code. We also thank Dr. Xiaoming Li, Chenyu Fan from the Molecular Imaging Core Facility (MICF) and Dr. Ying Xiong, Xiaoyue Ren from the Molecular Cellular Core Facility (MCCF) at the School of Life Sciences, ShanghaiTech University for providing technical support. This work was supported by the National Natural Science Foundation (NSFC) grant (32225023), Scientific & Technological Innovation 2030 Major Project of China (No. 2021ZD0203500 to J.-S.G.), NSFC grant (32130043) to H.X. and the Central Guidance on Local Science and Technology Development Fund (YDZX20233100001002).

## Author contributions

J.-S.G. and H.X. conceived, planned and supervised the project; X.G. designed, performed and analyzed the experiments in the paper with the help of P. H., S. X. and Y. Y; X.G and J.-S.G. wrote and revised the manuscript.

## Competing interests

The authors declare no competing interests.
