## [Peer Review File · Nature Communications]

Kdm4a is an activity downregulated barrier to generate engrams for memory separationREVIEWER COMMENTS

Reviewer #1 (Remarks to the Author):

The manuscript by Guo and colleagues investigates the role of epigenetic factors in the priming and formation of memory engrams in the hippocampus, a topic of high relevance. In particular, they identified histone demethylase Kdm4a as a novel negative regulator of engram formation by controlling the expression of a presynaptic protein Trpm7 via an mRNA binding protein Ythdc2, through sophisticated molecular strategies. The importance of these interactions was investigated in a contextual fear discrimination task.

To study the role of epigenetic factors involved in the generation of memory engrams, an *in vivo* CRISPR-Cas9 based gene knockout screen was performed in activated neurons isolated from the dentate gyrus (DG) of the EGR1-EGFP transgenic mice following a contextual fear conditioning task (CFC). Among all the candidates, Kdm4a emerged as a viable option for further study based on 'many'- but not shown- validation tests. The authors first showed that lentiviral shRNA mediated knockdown of Kdm4a in all DG neurons did not change the total number of activated neurons as assessed by neurons expressing two different activity markers Egr1 or cFos immunofluorescence. But, reducing Kdm4a expression increased the number of neurons recruited to CFC dependent activation as assessed by the number of EGFP+ cells co-expressing Egr1 or cFos. The authors also validated *in vitro* and *in vivo* reduction of Kdm4a mRNA levels upon neural activation as well as upon epileptic stimulation and developmentally not only in DG but also in all hippocampal subfields. From these experiments, it was interpreted that downregulated Kdm4a leads to increased 'allocation' of some activated neurons to the CFC engram. The authors then identified Trpm7 as a downstream target of Kdm4a, via recruitment by Ythdc2 and showed that shRNA mediated knockdown of Kdm4a or CRISPRa mediated increase of Trpm7 levels in DG granule cells increased size of the mossy fiber bouton without detectable changes in the dendrites of these cells. Finally, the authors validated their hypothesis that negative regulation of Kdm4a expression increases memory discrimination of different conditioning contexts in a manner similar to increasing the activity of this population. Conditional knockout of Kdm4a using a brain-wide Nestin-Cre driver line also leads to increased discrimination of non-shocked context without large effects on anxiety measures despite smaller size and lower body weight of the mutant mice.

The questions approached are important and the experimental design seems largely appropriate to address them. However, the rationale behind some of the experiments is confusing and the connection between results and conclusions is often unclear. Most importantly, authors are conflating memory allocation and discrimination. All the behavioral tasks presented in this manuscript test animals' ability to show differential recall (freezing responses) to a shocked (conditioned) context compared to a similar but non-conditioned context. The reason for similar freezing to Context A and B is likely due to the shared steel grid flooring for both A and B.

Typical tests for determining if the two engrams for Contexts A and C (no shared elements) are linked in time ('allocated') are done by exposing animals to these contexts prior to conditioning, separated by a couple of hours or a couple of days (for an example see Cai et al '16). This neutral exposure is followed by delivering an immediate shock in only one of these contexts- results in the transfer of memory from shocked to non-shocked contexts only if the initial, neutral exposures to these contexts occurred close in time. The Guo et al never tested this. Instead, the authors used a behavioral task that is based on exposure dependent discrimination of two similar contexts. This is sometimes termed as behavioral pattern separation and is critically dependent on DG mature and adult born granule cells. Hence the claim that Kdm4a downregulation is somehow important for 'decoupling contextual memories adjacent in time' should either be substantiated by additional experiments as described above or completely abandoned (lines 110, 114, 256, 260, 270, 272-274-as described above, CFD does not necessarily test linkage of adjacent memories-, 318, as well as discussion points). On the other hand, the authors present preliminary evidence for the importance of Kdm4a downregulation for

enhanced contextual discrimination ability (but see below).

A second caveat in this study is the EGR1-EGFP transgenic mouse line. The authors do not show low levels of EGFP expression before CFC experiments nor give any information about the half life of EGFP expression from EGR1 locus in their hands. How can they be certain about the fact that their FACS sorted cells represents engram neurons that represent the memory engrams for the CFC? I understand that they cannot repeat the whole study using one of the traditional inducible TRAP lines but they should at least validate the reduced mRNA or protein expression of Kdm4a in an inducible memory tagged mouse line or perform IEG co-expression analyses for Kdm4a levels.

A third issue, related to the second point above is the lack of cell type specificity in their viral and transgenic approaches. This also reduces the impact of their statements about the importance of the Kdm4a-Ythdc2-Trpm7 pathway for potentially regulating the outputs of DG granule cells involved in engram formation. For instance, in Figures 5 and 6, it is not clear whether the behavioral deficits arise from the broad impact of this pathway on all DG neurons or specifically neurons involved in engrams. Similarly, the authors showed the effect of increased neural activity in reducing Kdm4a levels in Figure 2 but not upon chemogenetic excitation of DG neurons (Figure 5 h-k), in which they present a behavioral phenotype. The crucial experiment showing reduced levels of Kdm4a, increased neural activity (or cells being potentially recruited to memory engrams) and facilitation of pattern separation all pointing to an interaction among these processes in the same cohort of mice is missing.

Additional concerns are listed below:

1. Some description regarding how wide 1 microliters of lentivirus spreads either in methods or with additional data is necessary to get a sense of whether/ how much the virus spread into CA3, hilar mossy cells or interneurons in DG/CA3. If so, the authors should consider this as a factor in their interpretation of lentiviral knockdown of Kdm4a across all components of this circuit and not just DG granule cells. There is a similar issue with AAV-hM3Dq-mCherry or control viruses used for experiments in Figure 5. The image in Extended Data Fig 9a shows strong expression of the viral mCherry in the hilus but weak expression in GCL. Additionally, this extends further into the lack of cell-type selectivity in their brain-wide Nestin-Cre transgenic strategy in Figure 6. Authors should discuss this as an experimental limitation together with its implications.
2. Methods used for Figures 4 and 5 should be thoroughly expanded, especially regarding quantitative analyses. For example, there are no descriptions of the staining, imaging or quantitative methods used for morphological analyses of DG GC dendrites (spine density, mushroom spines, spine head width) or size of mossy fiber boutons (methods for the entire Figure 4 is missing). There are also no descriptions of antibodies or details on confocal imaging or analysis of cell numbers regarding Egr1, cFos, EGFP, mCherry expression analyses.
3. In terms of behavioral analyses in figures 5 and 6, please describe pertinent details of the TSE apparatus and software for the sake of reproducibility. Were different cohorts used for each task, if not how many days in between, in which sequence? Many labs perform and analyze behavioral experiments differently and consequently sometimes get different results so it is important to give as detailed description as possible.
4. It will be useful to report the comparison of the initial percent freezing in Contexts A and B, instead of only reporting the discrimination index across days in Figures 5g and k, and 6j. There appears to be an immediate effect of Kdm4a knockdown in discrimination of A and B.
5. Please replace mCh with mCherry in figure 5k to match with figure legend.
6. The section of the Results in which authors describe the behavioral phenotype of Kdm4a KO mice should include the data presented in Extended Data Figure 10. Mice shown in Ext Fig 10b don't seem to be very different in size. Also, please clarify what shock speed means in Figure 6h- is this to measure somatosensation?
7. Please change wording in Line 294 '... significantly fired number of neurons in DG,..' to '..increased the activity of..'
8. The section between lines 251 and 260 and the corresponding section in the discussion should be rewritten. The reference cited for this section is on pyramidal neurons of visual cortex, which have

completely different input-output structure as well as intrinsic firing properties compared to DG granule cells. Granule cells do not display recurrent connectivity but have strong lateral inhibition as well as sparse activity resulting from a winner takes all process mediated by lateral inhibition. This reference and their explanation are not clearly relevant to what Figure 4 shows or how the Kdm4a-mediated presynaptic regulatory mechanism might contribute to the selective activation of neurons within an engram. Additionally, the link between the developmental role of Trpm7 and role of enlarged MF boutons for memory engrams is not clear.

9. For best impact authors should more carefully position their work in the context of conceptually related work on epigenetic modifications during memory trace formation (Jaeger et al, Nat Com '18, Marco et al, Nature '20 etc)

Reviewer #2 (Remarks to the Author):

The manuscript by Guo and colleagues aims to unravel the role of histone methylation-related genes in the allocation of engram cells participating in hippocampal-dependent memory tasks. By conducting a CRISPR-sgRNA-based screening targeting an extensive set of these proteins, the authors identified Kdm4a as a crucial epigenetic factor involved in the induction of activity-driven transcriptional programs. By using distinct molecular biology approaches, the authors identified the Trpm7 synaptic-plasticity related gene as a bona-fide target of Kdm4a protein, whose binding at particular Trpm7 genomic regions promotes transcriptional pausing and accumulation of Trpm7 partial transcripts to burst its expression in response to neuronal activity. In addition, the authors functionally characterize the relevance of these findings in the context of memory pattern separation tasks using chemogenetic approaches and generating a conditional Kdm4a KO mouse, providing great support to their molecular data and a plausible translational implication in memory-associated disorders. In general, the authors provide a remarkable set of data supporting their conclusions and deepen our understanding of epigenetic mechanisms contributing to memory allocation, which could be of great interest for the wide neuroscience community. However, the article contains some concerns that should be addressed before being considered for its publications in this journal:

Major points:

1- A major caveat in the experimental set-up of the CRISPR-sgRNA-based screening introduced in the first results section is that the cell populations sorted in the two different groups from which the experiment is performed do not have necessarily to correspond to the same cellular types, as EGFP would present a predominant neuronal expression due to the Egr1 promoter but mCherry and the Cas9 are expressed under the control of the ubiquitous EF1 α promoter. Could the authors rule out the possibility that the differences observed between mCherry+ EGFP- and mCherry + EGFP+ cells are not due to this distinct cellular composition? It is clear to me by observing the representative image in Extended Figure 1h that mCherry signal does not only label neuronal cells.

2- Observing the results presented in Figure 1C it is not obvious why the authors state that Kdm4a is the more robust and significant target from their CRISPR-sgRNA-based screening, having Kdm4d presenting a greater fold change and a higher MAGeCK p-value. It is important to understand the reasoning of these choice as the manuscript is built upon this decision. In this regard, Figure 2G

showing that Kdm4a is the only common gene identified among different sets of transcriptional datasets in the context of neuronal activity could be a supportive argument for focusing in Kdm4a if it would be shown in Figure 1 instead of Figure 2.

3- In line 92 the authors comment that with the results of their CRISPR-sgRNA-based screening they could determine "which epifactor knockout was able to attract the allocation of engram ensembles". I think at that point it is difficult to conclude with that experiment that the depletion of some of these epifactors can attract engram's allocation, but rather be dispensable for the cells to become part of the engram.

4- In line 104 the authors claim that "contextual fear conditioning task preferentially engaged activities in the neurons with lower Kdm4a expression". However, while a significant increase in Egr1+ cells can be observed when comparing shCtrl and shKdm4a, only 20% of Egr1 positive cells are presenting lower Kdm4a expression. Could the authors justify their statement?

5- In Figure 2f authors compare the levels of Egr1 and Kdm4a transcription in hippocampal DG of Egr1-GFP mice after CFC to show the transcriptional response of engram cells regarding these two particular genes. However, similarly to my previous comment, EFGP- cells could contain other cell types rather than just 'silent neurons' as stated in line 131. Could the authors evaluate this possibility by analyzing the level of distinct cell specific markers for glial and neuronal cells?

6- In line 185, the authors propose that Kdm4a recruitment to Trpm7 gene loci is mediated by Ythdc2 based on their PPI data, transcriptional correlation, and a localized nuclear pattern co-localization. While this data can indicate that the two proteins participate in common processes, it is not clear to me if it is proof enough to support the idea that Ythdc2 recruits Kdm4a to particular genomic loci.

7- Along the manuscript, molecular biology experiments are conducted indistinctly with different cell lines, including HEK293T, U2-OS and Neuro2A. Is there a particular reason for which that is performed in this way?

8- At statistical level, I have concerns with the number of replicates used for some analyses. For instance, some experiments are presented with N=2 and unpaired two-tailed t-test statistics are still applied (Figure 3i, Extended Data Fig. 3c) while in other cases the number of replicates doesn't seem to match with the number of samples shown in the graph, making difficult to know if the statistics are based on replicates or slices/samples (Extended Figure 2f for instance). Could the authors revise this?

Minor points:

1- In the results section, line 82, the authors comment that the dosage of lentivirus was adjusted to ensure that each infected neuron express a single sgRNA from the library, but no information of how this is achieved or validated is provided to support this. Could the authors elaborate a bit more regarding this?

2- I have found several typos along the manuscript that I think would require attention, including some phrases that are difficult to interpret or not adequately structured for a scientific article. I add some examples for the authors to check:

Line 30: "in many brain regions, that distinct cell..." -> "in many brain regions, where distinct cell..."

Line 47: "neuronal activities" -> "neuronal activity"

Line 58: "neuronal activities and track engram activities" -> "neuronal activity and track engram activity"

Line 103: "we designed shRNA targeting on Kdm4a" -> "we designed shRNA targeting Kdm4a"

Line 153: "changes via through RNA" -> "changes through RNA"

Line 179-182: Revise text

Line 228: "Fig. 3kand" -> "Fig. 3k and"

Line 234: "In the same time" -> "At the same time"

Line 285: "Therefore, Kdm4a-removal-mediated" -> "Therefore, Kdm4a-removal-mediated"

Line 334: "a priming state that neurons" -> "a priming state in which neurons"

Line 360: "It was known" -> "It is known"

Response to the reviewers' comments

We would like to thank the editor and reviewers for their positive feedback and helpful suggestions. These helpful comments and corrections give us an opportunity to update our study. To address the comments received, we have provided new experiments and corresponding data to the revised manuscript. We are also submitting the supplementary information and source data files, which includes both the raw and analyzed data. All modifications made to the text file of the manuscript have been highlighted in yellow for easy identification.

Our detailed responses to the reviewer's comments are as follows:

Reviewer #1 (Remarks to the Author):

The manuscript by Guo and colleagues investigates the role of epigenetic factors in the priming and formation of memory engrams in the hippocampus, a topic of high relevance. In particular, they identified histone demethylase Kdm4a as a novel negative regulator of engram formation by controlling the expression of a presynaptic protein Trpm7 via an mRNA binding protein Ythdc2, through sophisticated molecular strategies. The importance of these interactions was investigated in a contextual fear discrimination task.

Thanks for the evaluation and insightful comments.

To study the role of epigenetic factors involved in the generation of memory engrams, an in vivo CRISPR-Cas9 based gene knockout screen was performed in activated neurons isolated from the dentate gyrus (DG) of the EGFR1-EGFP transgenic mice following a contextual fear conditioning task (CFC). Among all the candidates, Kdm4a emerged as a viable option for further study based on 'many'- but not shown- validation tests.

The authors first showed that lentiviral shRNA mediated knockdown of Kdm4a in all DG neurons did not change the total number of activated neurons as assessed by neurons expressing two different activity markers Egr1 or cFos immunofluorescence. But, reducing Kdm4a expression increased the number of neurons recruited to CFC dependent activation as assessed by the number of EGFP+ cells co-expressing Egr1 or cFos. The authors also validated in vitro and in vivo reduction of Kdm4a mRNA levels upon neural activation as well as upon epileptic stimulation and developmentally not only in DG but also in all hippocampal subfields. From these experiments, it was interpreted that downregulated Kdm4a leads to increased 'allocation' of some activated neurons to the CFC engram. The authors then identified Trpm7 as a downstream target of Kdm4a, via recruitment by Ythdc2 and showed that shRNA mediated knockdown of Kdm4a or CRISPRa mediated increase of Trpm7 levels in DG granule cells increased size of the mossy fiber bouton without detectable changes in the dendrites of these cells. Finally, the authors validated their hypothesis that negative regulation of Kdm4a expression increases memory discrimination of different conditioning contexts in a manner similar to increasing the activity of this population. Conditional knockout of Kdm4a using a brain-wide Nestin-Cre driver line also leads to increased discrimination of non-shocked context without large effects on anxiety measures despite smaller size and lower body weight of the

mutant mice.

The questions approached are important and the experimental design seems largely appropriate to address them.

Thanks for the overall evaluation to the importance of our study and our efforts.

However, the rationale behind some of the experiments is confusing and the connection between results and conclusions is often unclear. Most importantly, authors are conflating memory allocation and discrimination. All the behavioral tasks presented in this manuscript test animals' ability to show differential recall (freezing responses) to a shocked (conditioned) context compared to a similar but non-conditioned context.

Thanks for pointing out the important question in our study. We now have completed new experiments to address those questions and have revised the manuscript.

The reason for similar freezing to Context A and B is likely due to the shared steel grid flooring for both A and B.

Typical tests for determining if the two engrams for Contexts A and C (no shared elements) are linked in time ('allocated') are done by exposing animals to these contexts prior to conditioning, separated by a couple of hours or a couple of days (for an example see Cai et al '16). This neutral exposure is followed by delivering an immediate shock in only one of these contexts- results in the transfer of memory from shocked to non-shocked contexts only if the initial, neutral exposures to these contexts occurred close in time. The Guo et al never tested this.

*Instead, the authors used a behavioral task that is based on exposure dependent discrimination of two similar contexts. This is sometimes termed as behavioral pattern separation and is critically dependent on DG mature and adult born granule cells. Hence the claim that *Kdm4a* downregulation is somehow important for 'decoupling contextual memories adjacent in time' should either be substantiated by additional experiments as described above or completely abandoned (lines 110, 114, 256, 260, 270, 272-274-as described above, CFD does not necessarily test linkage of adjacent memories-, 318, as well as discussion points).*

To directly answer this question, we carried out a new set of experiments as suggested by the reviewer. We performed the memory allocation test to determine whether artificially decreasing *Kdm4a* levels in a subset of DG neurons affects the ability to allocate contextual memories that occurred close in time. Mice injected with LV-U6-sh*Kdm4a* or control virus were subjected to a typical memory allocation test¹. Briefly, the mice were subjected to freely explore three distinct, novel contexts which were separated by 7 days (context C and context B) or 5 h (context A and context B). Two days later, mice were placed in the context A and given an immediate footshock (**Response to Reviewer Figure 1a**). Two days after the footshock, the mice were tested in either context A (shocked context), B (5 h; not shocked), or C (7days; not shocked). As previously reported¹, the control mice (shCtrl) froze similarly in context A and B, exhibited memory-linking between distinct memories encoded close in time. However, the *Kdm4a*-knockdown mice displayed a significantly lower freezing in context B than that in context A (**Response to Reviewer Figure 1b**), which suggests

that downregulation of *Kdm4a* in DG might have impact on memory allocation to separate linked memories closed in time.

Response to Reviewer Figure 1. (also see Fig. 5l-m, Extended Data Fig. 13d)

a. Characterization of the memory allocation test and context (Imm shock, immediate shock). **b.** *Kdm4a*-knockdown mice froze significantly lower in context B than that in context A.

It is noticed that for the discrimination, context A, B and C did not share the same floor as shown in **Response to Reviewer Figure 1a (right)**. These results remind us the CREB-mediated memory allocation mechanism that eligible neurons are selected to participate in a memory trace as a function of their relative CREB activity at the time of learning². When neutral exposure of context A and B are close in time, so that context B induced CREB-p remains active when experiencing context A, those memories are allocated to the same neuron ensembles, showing as memory linking. In our study, decreasing of *Kdm4a* in DG neurons also selected neurons to participate in memory traces (**Fig.1**), similar to the function of increasing CREB-p. Therefore, knockdown of *Kdm4a* in a subset of neurons plays an independent process of CREB-activity triggered neuronal selection of memory traces (memory linking) to allocate memories of time-closed contexts in different ensembles. In one hand, our study supports the original discovery of memory linking³, on the other hand, our study suggests epigenetic mechanisms also play a role in memory trace allocation to regulate the memory linking phenomenon.

We appreciate the opportunity here to add new experiments to clarify the role of *Kdm4a* in memory allocation. We performed the classical memory allocation test as suggested above, and confirmed that *Kdm4a* downregulation in the DG affect the linked memories close in time. In the revised manuscript, we addressed issue of pattern separation experiments and the memory allocation experiments.

*On the other hand, the authors present preliminary evidence for the importance of *Kdm4a* downregulation for enhanced contextual discrimination ability (but see below).*

Thanks for this point. We have performed new experiments to increase the evidence for enhanced contextual discrimination ability. Previous studies^{1,4} showed that fear memory formation increases the neuronal excitability in hippocampal neurons at 5 h post context exposure. Therefore, we assessed the freezing levels of *Nestin-Cre*-mediated *Kdm4a* cKO mice and control mice (*Kdm4a^{fl/fl}*) at 5 h post the fear acquisition. Briefly, mice were subjected to two distinct, novel contexts (context A and context B,

different floors). Mice were given an immediate footshock in context A, then returned to the homecage. 5 h later mice were placed in context B for another 3 min of exploration (**Response to Reviewer Figure 2a**). In the control mice there was no difference between freezing in context A and context B. Remarkably, we found that the *Kdm4a* cKO mice tested in context B froze significantly less than that in the context A, indicating that *Kdm4a* downregulation increased the discrimination abilities toward distinct contexts (**Response to Reviewer Figure 2b**).

Response to Reviewer Figure 2. (also see Extended Data Fig. 13e-f)

a. Schematics of a task for exploration of two different contexts close in time. **b.** *Kdm4a* cKO mice exhibit lower freezing levels in B compared to A at 5 hours after immediate shock.

A second caveat in this study is the EGR1-EGFP transgenic mouse line. The authors do not show low levels of EGFP expression before CFC experiments nor give any information about the half life of EGFP expression from EGR1 locus in their hands. How can they be certain about the fact that their FACS sorted cells represents engram neurons that represent the memory engrams for the CFC?

Our previous studies used *Egr1-EGFP* transgenic mouse to monitor the stimulus-induced neuronal activities in the cortex⁵⁻⁷ and hippocampus^{8,9}. The exogenous EGFP mRNA levels of *Egr1-EGFP* mouse were linearly correlated with the endogenous *Egr1* mRNA levels in neurons. *Egr1* and EGFP protein levels were significantly increased in *Egr1-EGFP* mice 1.5-2 h after injection of kainic acid (20 mg/kg), and then returned to the basal levels within 3-6 h⁷. To validate the memory-associated EGFP levels of *Egr1-EGFP* transgenic line, mice were subjected to the contextual fear conditioning (CFC) and then returned to the homecage. We detected that both the number of EGFP+ cells and EGFP fluorescence intensity significantly increased in the DG granule cell layers (GCL) at 1.5 h post CFC (**Response to Reviewer Figure 3a-c**).

Additionally, we also detected both the CFC-induced increase of *Egr1* and EGFP mRNA levels in the FACS sorted EGFP+ cells (**Fig. 2f, Extended Data Fig. 6h-j**) at 1.5 h post CFC. Please also see the new experiments for details (**Response to Reviewer Figure 9**). Taken together, the CFC-driven expression of EGFP can be used to identify the engaged engram ensembles in the *Egr1-EGFP* transgenic mouse.

Response to Reviewer Figure 3. (also see Extended Data Fig. 1g-i)

a, Representative images of Egr1-EGFP-expressing DG GCs after CFC. **b,** Quantification of EGFP positive cell number in homecage or CFC. **c,** Quantification of EGFP fluorescence intensity in homecage or CFC.

*I understand that they cannot repeat the whole study using one of the traditional inducible TRAP lines but they should at least validate the reduced mRNA or protein expression of *Kdm4a* in an inducible memory tagged mouse line or perform IEG co-expression analyses for *Kdm4a* levels.*

To directly answer this question, we performed fluorescence activated nuclei sorting (FANS) to detect the fear memory-associated *Kdm4a* levels in Immediate-early gene (IEG)-expressed DG GCs. Mice were subjected to the CFC, returned to the homecage, and then perfused after 1 h. Individual nuclei were isolated from DG GCL using Dounce homogenization. To identify the fear memory-activated DG GCs, nuclei were co-immunostained with NeuN and Fos antibodies. DG GC nuclei were identified using Hoechst-33342 via flow sorting (**Response to Reviewer Figure 4a**). We observed 2.39% of activated neuronal population (NeuN+Fos+) in the DG GC nuclei (**Response to Reviewer Figure 4b-c**). The NeuN+Fos+ and NeuN+Fos- nuclei were directly sorted into 0.2-ml centrifuge tubes and amplified using the Smart-seq2 protocol¹⁰.

Real time qPCR assay was performed to measure the mRNA expression levels of *Fos* and *Kdm4a*. As expected, we detected the mRNA expression of *Kdm4a* was decreased in NeuN+Fos+ nuclei population compared to NeuN+Fos- nuclei (**Response to Reviewer Figure 4d**), thereby, confirming our original discovery of decreased *Kdm4a* level in activated DG neurons. In addition, we examined the difference in cell type-specific gene expression between NeuN+Fos+ and NeuN+Fos- nuclei and confirmed that these two FANS sorted nuclei were homogeneous DG GC populations without the perturbation resulted from non-neuronal cells (**Response to Reviewer Figure 4e**).

Response to Reviewer Figure 4. (also see Extended Data Fig. 7)

a, Schematics of flow sorting strategy for the isolation of fear memory-associated nuclei from the DG. **b**, Representative FANS plots showing the gating strategy for the identification of NeuN+Fos+ nuclei in the DG. **c**, Representative images of FANS sorted nuclei. **d**, Fos (left) and Kdm4a (right) mRNA expression levels in the NeuN+Fos- or NeuN+Fos+ nuclei. **e**, Several marker gene mRNA expression levels in the NeuN+Fos- or NeuN+Fos+ nuclei. (*Slc17a7* (excitatory neuron marker), *Gad1* (inhibitory neuron marker), *Aif1* (microglia marker), *Plp1* (Oligodendrocyte cell marker), *Pdgfra* (Oligodendrocyte precursor cell marker), *Aldh111* (astrocyte marker), and *Flt1* (endothelial cell marker)).

A third issue, related to the second point above is the lack of cell type specificity in their viral and transgenic approaches. This also reduces the impact of their statements about the importance of the Kdm4a-Ythdc2-Trpm7 pathway for potentially regulating the outputs of DG granule cells involved in engram formation.

For instance, in Figures 5 and 6, it is not clear whether the behavioral deficits arise from the broad impact of this pathway on all DG neurons or specifically neurons involved in engrams.

As expected by the reviewer, we have added new analysis on the cell type specificity in a series of new experiments (See in **Response to Reviewer Figure 4, 7, 9**). It appears that a subpopulation of dentate neurons with reduced Kdm4a expression contribute to the behavior phenotypes of increased discrimination between contextual memories closed in time. In those assays, only a proportion of dentate neurons showed Kdm4a knockdown in Figure 5. As discussed in **Response to Reviewer Figure 11**, only a portion of Kdm4a knockdown neurons are activated during contextual memory tasks. This allows the facilitation of memory allocation to discriminate memories close in time, as shown in **Response to Reviewer Figure 1**. Thereby, we speculate that only a subpopulation of dentate neurons in engrams but not all DG neurons contribute in the behavioral phenotype.

Similarly, the authors showed the effect of increased neural activity in reducing Kdm4a levels in Figure 2 but not upon chemogenetic excitation of DG neurons (Figure 5 h-k), in which they

present a behavioral phenotype.

The crucial experiment showing reduced levels of *Kdm4a*, increased neural activity (or cells being potentially recruited to memory engrams) and facilitation of pattern separation all pointing to an interaction among these processes in the same cohort of mice is missing.

To directly answer this question, we measured the *Kdm4a* mRNA expression levels in the CNO injected hM3Dq mice. Consistent with the prior experiments, chemogenetic activation of DG GCs similarly induced the decrease of endogenous *Kdm4a* mRNA expression levels *in vivo* (**Response to Reviewer Figure 5a, b**).

Response to Reviewer Figure 5. (also see Extended Data Fig. 13e)

a, Schematics of chemogenetic activation of DG GCs. **b**, *Fos* (left), *Egr1* (middle) and *Kdm4a* (right) mRNA expression levels were measured by RT-qPCR.

In addition to this experiment, we have also performed the contextual fear conditioning task and isolated the active neurons. We identified that context-activated neuron ensembles showed increased IEG-expression and decreased *Kdm4a* expression (**Response to Reviewer Figure 4**).

Additional concerns are listed below:

1. Some description regarding how wide 1 microliters of lentivirus spreads either in methods or with additional data is necessary to get a sense of whether/ how much the virus spread into CA3, hilar mossy cells or interneurons in DG/CA3. If so, the authors should consider this as a factor in their interpretation of lentiviral knockdown of *Kdm4a* across all components of this circuit and not just DG granule cells.

There is a similar issue with AAV-hM3Dq-mCherry or control viruses used for experiments in Figure 5. The image in Extended Data Fig 9a shows strong expression of the viral mCherry in the hilus but weak expression in GCL. Additionally, this extends further into the lack of cell-type selectivity in their brain-wide Nestin-Cre transgenic strategy in Figure 6. Authors should discuss this as an experimental limitation together with its implications.

As suggested by the reviewer we have performed a quantification of lentiviral spread and multiplicity of infection in the DG and neighboring regions (e.g. Hilus and CA3). Firstly, to validate the lentiviral spread, we injected the LV-U6-sgCtrl-Ef1 α -dCas9-P2A-mCherry lentivirus into the dentate gyrus of adult C57BL/6J mice using the same titer and injection parameters as used for the sgRNA lentiviral library. Serial 40 μ m sections through the hippocampus were cut and stained for mCherry antibody and DAPI. The staining results indicated that 1 microliter of lentivirus would spread roughly the whole

dorsal DG (from AP-1.0 to AP -3.0). The upper blade of ventral DG was partially infected because the limitation of lentiviral spread (**Response to Reviewer Figure 6b, AP-3.2**).

Response to Reviewer Figure 6. (also see Extended Data Fig. 2)

a. Schematics of delivery of lentivirus into the DG. **b.** Representative images of serial sections of LV-injected mouse brain.

Secondly, we determined that the lentivirus primarily infected the granule cells (91.53%) in the GCL, and only a very small population of LV-infected neuronal cells were detected in the Hilus (4.53%) and partial CA3 (3.94%) between the upper and lower blade of the DG (**Response to Reviewer Figure 7a-c**). Additionally, we stained LV-injected tissues with the GABAergic interneuron (INs) makers Parvalbumin (PV) or Somatostatin (SST), and observed that the percentage of double stained cells are both very small (<2%) due to the nature of sparsity of PV and SST neurons in the DG (**Response to Reviewer Figure 7d-g**). Lentivirus could infect non-neuronal cells in the DG and expressed mCherry proteins because the ubiquitous EF1 α promoter, to address this possibility, we stained LV-injected tissues with several markers of glial cells (GFAP, astrocyte marker; Olig2, oligodendrocyte marker; Iba1, microglia marker) and observed a few of co-stained cells among GFAP (~3%), Olig2 (~1%), Iba1 (0%) and mCherry positive cells in the DG (**Response to Reviewer Figure 7i-l**). While the NeuN immunostaining showed mCherry signals were predominantly enriched in the neuronal cells in the DG (**Response to Reviewer Figure 7h, i**). Thus, these evidences point to the conclusion that the vast majority of mCherry-expressing neurons in the DG are excitatory granule cells (>90%).

Moreover, our immunostaining results revealed that AAV-CaMKII α -hM3Dq-mCherry virus predominantly infected the excitatory neurons in the DG. However, owing to the injection site in the hilus, a minority of neurons in the hilus and CA3 were inevitably infected as well. To validate this, we replicated the AAV injection experiment, which confirmed that the infection was primarily confined to the DG (**Extended Data Fig. 13**). Furthermore, we acknowledged this as a limitation in our experimental design within

the manuscript.

Response to Reviewer Figure 7. (also see Extended Data Fig. 3)

a, Representative images of co-staining of mCherry and NeuN in the GCL, Hilus and CA3. **b**, The number of mCherry+NeuN+ double positive cells in the GCL, Hilus and CA3. **c**, The percentage of mCherry+NeuN+ double positive cells in the GCL, Hilus and CA3. **d**, Representative images of co-staining of mCherry and PV+ INs. **e**, Representative images of co-staining of mCherry and SST+ INs. **f**, The number of INs in the DG. **g**, The percentage of double stained cells of mCherry and different interneuron markers. **h**, Representative images of co-staining of mCherry and NeuN. **i**, Representative images of co-staining of mCherry and GFAP. **j**, Representative images of co-staining of mCherry and Olig2. **k**, Representative images of co-staining of mCherry and Iba1. **l**, The percentage of double stained

cells of mCherry and different cell markers.

2. *Methods used for Figures 4 and 5 should be thoroughly expanded, especially regarding quantitative analyses. For example, there are no descriptions of the staining, imaging or quantitative methods used for morphological analyses of DG GC dendrites (spine density, mushroom spines, spine head width) or size of mossy fiber boutons (methods for the entire Figure 4 is missing). There are also no descriptions of antibodies or details on confocal imaging or analysis of cell numbers regarding Egr1, cFos, EGFP, mCherry expression analyses.*

Thank you for reminding us. As suggested by the reviewer we have expanded the details and procedures of imaging analysis in our Methods part.

3. *In terms of behavioral analyses in figures 5 and 6, please describe pertinent details of the TSE apparatus and software for the sake of reproducibility. Were different cohorts used for each task, if not how many days in between, in which sequence? Many labs perform and analyze behavioral experiments differently and consequently sometimes get different results so it is important to give as detailed description as possible.*

We appreciate this concern. We have used TSE apparatus following the same protocol from 2014^{5,6,8,9,11}. As suggested by the reviewer we have updated our animal behavior protocols, and detailly described the apparatus and software used in our Methods (Behavioral analysis) part.

4. *It will be useful to report the comparison of the initial percent freezing in Contexts A and B, instead of only reporting the discrimination index across days in Figures 5g and k, and 6j. There appears to be an immediate effect of Kdm4a knockdown in discrimination of A and B.*

Here we plot the data of first day and day 4 for the three sets of experiments (**Response to Reviewer Figure 5**). In addition, we have added new experiments of immediate shock in Context A, then tested in context A or context B 5 hours later to ask if the discrimination ability is altered. We did see the Kdm4a knockdown group showed discrimination of A and B (**Response to Reviewer figure 2**). We now added discussion on its effect in discrimination.

Response to Reviewer Figure 2. (also see Extended Data Fig. 13e-f)

a. Schematics of a task for exploration of two different contexts close in time. b. Kdm4a cKO mice exhibit lower freezing levels in B compared to A at 5 hours after immediate shock.

Response to Reviewer Figure 8.

a-b & e-f data for figure 5. c-d data for figure 6

5. Please replace mCh with mCherry in figure 5k to match with figure legend.

We agree with the reviewer and have replaced mCh with mCherry in Figure 5k to match the figure legend.

6. The section of the Results in which authors describe the behavioral phenotype of Kdm4a KO mice should include the data presented in Extended Data Figure 10. Mice shown in Ext Fig 10b don't seem to be very different in size. Also, please clarify what shock speed means in Figure 6h- is this to measure somatosensation?

The body weight of Kdm4a cKO mice is reduced by approximately 10% (**Extended Data Fig. 14b-c**), whereas their body size did not show significant reduction. We have refined the title of the figure legend for greater clarity. The shock speed is related to their sensation to the electrical pulses. As a control experiment, it indicated that the Kdm4a cKO mice exhibited normal responses in training section when compared with the control group. This clarification has been included in the main text.

7. Please change wording in Line 294 '... significantly fired number of neurons in DG,..' to '...increased the activity of..'

We agree with the reviewer and have now changed the wording to clearly express that the results of chemogenetic excitation of neurons.

8. The section between lines 251 and 260 and the corresponding section in the discussion should be rewritten. The reference cited for this section is on pyramidal neurons of visual cortex, which have completely different input-output structure as well as intrinsic firing properties compared to DG granule cells. Granule cells do not display recurrent connectivity but have strong lateral inhibition as well as sparse activity resulting from a winner takes all process mediated by lateral inhibition. This reference and their explanation are not clearly relevant to what Figure 4 shows or how the Kdm4a-mediated presynaptic regulatory mechanism might contribute to the selective activation of neurons within an engram.

Additionally, the link between the developmental role of Trpm7 and role of enlarged MF boutons for memory engrams is not clear.

We have modified the description here. Trpm7 has been reported to regulate synaptic activities^{12,13}. Here we observed that Trpm7 is related to Kdm4a in the enlarged MF boutons and might contribute to the interaction between memory engrams to other cells. In the lab, we now started a new program to answer this question directly to investigate the interaction between granule cell and local inhibitory neurons in DG. Unpublished data showed that synapses of Kdm4a knockdown DG neurons might have stronger synaptic outputs to the inhibitory neuron, so as to induce higher lateral inhibition to others.

9. For best impact authors should more carefully position their work in the context of conceptually related work on epigenetic modifications during memory trace formation (Jaeger et al, Nat Com '18, Marco et al, Nature '20 etc)

We appreciate your reminder. We have cited those papers and discussed our studies with those conceptually related works in the introduction and discussion section.

Reviewer #2 (Remarks to the Author):

The manuscript by Guo and colleagues aims to unravel the role of histone methylation-related genes in the allocation of engram cells participating in hippocampal-dependent memory tasks. By conducting a CRISPR-sgRNA-based screening targeting an extensive set of these proteins, the authors identified Kdm4a as a crucial epigenetic factor involved in the induction of activity-driven transcriptional programs. By using distinct molecular biology approaches, the authors identified the Trpm7 synaptic-plasticity related gene as a bona-fide target of Kdm4a protein, whose binding at particular Trpm7 genomic regions promotes transcriptional pausing and accumulation of Trpm7 partial transcripts to burst its expression in response to neuronal activity. In addition, the authors functionally characterize the relevance of these findings in the context of memory pattern separation tasks using chemogenetic approaches and generating a conditional Kdm4a KO mouse, providing great support to their molecular data and a plausible translational implication in memory-associated disorders.

In general, the authors provide a remarkable set of data supporting their conclusions and deepen our understanding of epigenetic mechanisms contributing to memory allocation, which could be of great interest for the wide neuroscience community.

Thanks for the valuable comments on our study. We sincerely appreciate the recommendation on our work by this reviewer. In response to the comments received, we have provided new experiments and corresponding data to the revised manuscript.

However, the article contains some concerns that should be addressed before being considered for its publications in this journal:

Major points:

1- A major caveat in the experimental set-up of the CRISPR-sgRNA-based screening introduced in the first results section is that the cell populations sorted in the two different groups from which the experiment is performed do not have necessarily to correspond to the same cellular types, as EGFP would present a predominant neuronal expression due to the Egr1 promoter but mCherry and the Cas9 are expressed under the control of the ubiquitous EF1 α promoter. Could the authors rule out the possibility that the differences observed between mCherry+ EGFP- and mCherry + EGFP+ cells are not due to this distinct cellular composition? It is clear to me by observing the representative image in Extended Figure 1h that mCherry signal does not only label neuronal cells.

In the immunostaining experiment described above, we found that the vast majority of the mCherry-expressing neurons are excitatory granule cells in the DG (>90%), only a small portion of mCherry positive non-neuronal cells were detected in the LV-infected tissues (**Response to Reviewer Figure 7h-I**).

To directly address his point, we examined the cell type specificity of mCherry positive cells that sorted from the LV-injected *Egr1-EGFP* mice (**Response to Reviewer Figure 9a**). The lentivirus harboring the mCherry cDNA was delivered into the DG of adult *Egr1-EGFP* mice. Mice were subjected to a CFC task, and perfused after 1.5 h.

Single cells were isolated from the DG through papain-based enzymatic dissociation. The mCherry-expressing cells were identified using a flow cytometry, and then sorted directly into 0.2-ml centrifuge tubes containing the lysis buffer (**Response to Reviewer Figure 9b-d**). RNA from the mCherry+EGFP- and mCherry+EGFP+ cells were amplified and reverse transcribed using the Smart-seq2 protocol¹⁰. Firstly, we confirmed that the purity of sorted mCherry+ cells by measuring the mCherry mRNA expressions in FACS-sorted cells (**Response to Reviewer Figure 9e**), and detected the increased *EGFP*, *Egr1*, and decreased *Kdm4a* mRNA levels in the mCherry+EGFP+ cells compared to the mCherry+EGFP- cells due to the neuronal activation. Secondly, we measured several cell-type specific gene expression in FACS-sorted cells and found there were no difference between the mCherry+EGFP- and mCherry+EGFP+ cells in *Slc17a7* (excitatory neuron marker), *Gad1* (inhibitory neuron marker), *Aif1* (microglia marker), *Plp1* (Oligodendrocyte cell marker), *Pdgfra* (Oligodendrocyte precursor cell marker), *Aldh111* (astrocyte marker), and *Flt1* (endothelial cell marker) mRNA expression levels (**Response to Reviewer Figure 9f**). Thus, the differences that we observed between the mCherry+EGFP- and the mCherry+EGFP+ cells in this study are not due to distinct cellular composition.

Response to Reviewer Figure 9. (also see Extended Data Fig. 4)

a, Schematics of the isolation of lentiviral infected cells from the DG. **b-c**, Representative FACS plots showing the gating strategy for the identification of mCherry-expressing cells in the DG. **d**, Representative images of FACS sorted mCherry+EGFP+ cell. **e**, Expression levels of *mCherry*, *EGFP*, *Egr1* and *Kdm4a* mRNA in the FACS sorted cells. **f**,

Cell-type specific gene expression in the FACS sorted cells.

2- Observing the results presented in Figure 1C it is not obvious why the authors state that *Kdm4a* is the more robust and significant target from their CRISPR-sgRNA-based screening, having *Kdm4d* presenting a greater fold change and a higher MAGeCK *p*-value. It is important to understand the reasoning of these choice as the manuscript is built upon this decision. In this regard, Figure 2G showing that *Kdm4a* is the only common gene identified among different sets of transcriptional datasets in the context of neuronal activity could be a supportive argument for focusing in *Kdm4a* if it would be shown in Figure 1 instead of Figure 2.

Thanks for your kind reminder. We agree with the reviewer that Figure 2G should be shown in Figure 1 and served as a supportive evidence for focusing in *Kdm4a*. During the analysis of the screening results, we found some enriched genes which present high fold change and robust MAGeCK *p*-value were caused by the biased loss of sgRNAs after *in vivo* screening, e.g. *Kdm4d* sgRNAs (**Response to Reviewer Figure 10a**). The normalized sgRNA counts have no difference between mCherry+EGFP- and mCherry+EGFP+ cell population (**Fig. 1d**). Furthermore, a previous transcriptome study¹⁴ showed that the expression of *Kdm4d* represent a relatively low level in the mouse brain when compared with other *Kdm4* family genes (**Response to Reviewer Figure 10b**), which is also confirmed by the *Kdm4d* RNA *in situ* hybridization data from the Allen brain atlas (**Response to Reviewer Figure 10c**). Additionally, as you suggested above, *Kdm4a* is the only common gene identified among different activity-downregulated transcriptional datasets. Based on these evidences, we exclude *Kdm4d* from genes of interest.

Response to Reviewer Figure 10. (also see Fig. 1d)

a, Normalized counts of *Kdm4d* sgRNAs in the mCherry+EGFP- and mCherry+EGFP+ cell population. **b**, The FPKM count of *Kdm4* family genes in the adult mouse brain, from ref¹⁴. **c**, *In situ* hybridization data of *Kdm4* family genes in the adult mouse brain.

3- In line 92 the authors comment that with the results of their CRISPR-sgRNA-based screening they could determine “which epifactor knockout was able to attract the allocation of engram ensembles”. I think at that point it is difficult to conclude with that experiment that the depletion of some of these epifactors can attract engram’s allocation, but rather be dispensable for the cells to become part of the engram.

We appreciate for your suggestions and we have rewritten this sentence to better

express the results of the screening.

4- In line 104 the authors claim that “contextual fear conditioning task preferentially engaged activities in the neurons with lower *Kdm4a* expression”. However, while a significant increase in *Egr1*⁺ cells can be observed when comparing *shCtrl* and *shKdm4a*, only 20% of *Egr1* positive cells are presenting lower *Kdm4a* expression. Could the authors justify their statement?

In light of this comment, we now have put *Egr1*⁺ and *Fos*⁺ cell both in the quantification (**Response to Reviewer Figure 11**). Both of those data suggest a significant increase of abundance of task-activated neuron population in the *shKdm4a* group. Although only 20% *Egr1*⁺ neurons are in the *shKdm4a* population, it was significantly higher than the 5% *Fos*⁺ in *shCtrl* group and 10% *Egr1*⁺ in *shCtrl* group. 80% of *Egr1*⁺ in *EYFP*⁻ neurons did not exclude the role of *Kdm4a* in regulating the distribution of engaged memory traces, as the population of *Egr1*⁺*EYFP*⁻ neurons did not exclude the *Kdm4a* downregulation by developmental and environmental factors (prior to the fear conditioning task) in this population. Moreover, as the strong lateral inhibition in the DG, very few neurons are activated during contextual fear conditioning, so that not all the virus-expressing *shKdm4a* neurons can be activated. Thus, the virus-expressing of *shKdm4a* only increased the preference of memory allocation in the *EYFP*⁺ cell population. Here we modified our statement in the main text.

Response to Reviewer Figure 11. (also see Fig. 1k and Extended Data Fig. 5g)

a, Quantification of *Egr1* immunostaining (white bars: *Egr1*⁺*EYFP*⁻ cells; green bars: *Egr1*⁺*EYFP*⁺ cells). **b**, Quantification of *Fos* immunostaining (white bars: *Fos*⁺*EYFP*⁻ cells; orange bars: *Fos*⁺*EYFP*⁺ cells).

5- In Figure 2f authors compare the levels of *Egr1* and *Kdm4a* transcription in hippocampal DG of *Egr1*-GFP mice after CFC to show the transcriptional response of engram cells regarding these two particular genes. However, similarly to my previous comment, *EYFP*⁻ cells could contain other cell types rather than just ‘silent neurons’ as stated in line 131. Could the authors evaluate this possibility by analyzing the level of distinct cell specific markers for glial and neuronal cells?

It is an important question. We performed RT-qPCR analysis of cell type composition in the FACS-sorted *EGFP*⁻ and *EGFP*⁺ cells (**Response to Reviewer Figure 12a**). The *Egr1*-*EGFP* transgenic mice were subjected to explore a CFC task, and perfused after 1.5 h. The dissociated single cells of the DG were sorted and identified using a flow cytometry (**Response to Reviewer Figure 12b-c**). The *EGFP*⁻ cells and the *EGFP*⁺ cells were collected for downstream analysis. RT-qPCR analysis confirmed that the activity-dependent transcription of *EGFP* and *Egr1* mRNA is predominantly

enriched in the EGFP+ cells compared to the EGFP- cells, which is also validated by the activity-induced downregulation of *Kdm4a* expressions (**Response to Reviewer Figure 12d**).

However, we observed that *Slc17a7* (excitatory neuron marker) and *Aif1* (microglia marker) mRNA levels were greater in the EGFP+ cells than in the EGFP- cells, and the *Plp1* (Oligodendrocyte cell marker), *Pdgfra* (Oligodendrocyte precursor cell marker), *Aldh11* (astrocyte marker), and *Flt1* (endothelial cell marker) mRNA levels were increased in the EGFP- cells when compared to those in the EGFP+ cells. These results indicated that, as expected, the FACS-sorted EGFP- cells contain glial cells. However, the amount of the glia-marker expression in EGFP- negative cells were not high. We expect the main contribution of activity-induced downregulation of *Kdm4a* came from neuronal population.

To further justify our conclusion, we performed the fluorescence activated nuclei sorting (FANS) to detect the fear memory-associated *Kdm4a* levels in the Fos positive nuclei (**Response to Reviewer Figure 4a**). The neuronal cells were identified using NeuN immunostaining. RT-qPCR analysis confirmed that cell-type specific gene expression has no difference between the FANS-sorted NeuN+Fos- and NeuN+Fos+ nuclei (**Response to Reviewer Figure 4e**). Consistent with our prior experiments, we detected the decreased *Kdm4a* mRNA expression levels in the memory-associated NeuN+Fos+ nuclei population compared to the nonactivated NeuN+Fos- nuclei (**Response to Reviewer Figure 4d**).

Response to Reviewer Figure 4. (also see Extended Data Fig. 7)

a. Schematics of flow sorting strategy for the isolation of fear memory-associated nuclei from the DG. **b.** Representative FANS plots showing the gating strategy for the identification of NeuN+Fos+ nuclei in the DG. **c.** Representative images of FANS sorted nuclei. **d.** Fos (left) and *Kdm4a* (right) mRNA expression levels in the NeuN+Fos- or NeuN+Fos+ nuclei. **e.** Several marker gene mRNA expression levels in the NeuN+Fos- or NeuN+Fos+ nuclei. (*Slc17a7* (excitatory neuron marker), *Gad1* (inhibitory neuron marker), *Aif1* (microglia marker), *Plp1* (Oligodendrocyte cell marker), *Pdgfra*

(Oligodendrocyte precursor cell marker), *Aldh111* (astrocyte marker), and *Flt1* (endothelial cell marker)).

Response to Reviewer Figure 12. (also see Fig. 2f and Extended Data Fig. 6h-j)

a, Schematics of the isolation of the CFC-activated EGFP+ neurons from the DG of *Egr1-EGFP* transgenic mice. **b**, Representative FACS plots showing the gating strategy for the identification of EGFP-expressing cells in the DG. **c**, Representative images of FACS sorted EGFP- cell and EGFP+ cell. **d**, Expression levels of *EGFP*, *Egr1* and *Kdm4a* mRNA in the FACS sorted cells. **e**, Cell-type specific gene expression in the FACS sorted cells.

6- In line 185, the authors propose that *Kdm4a* recruitment to *Trpm7* gene loci is mediated by *Ythdc2* based on their PPI data, transcriptional correlation, and a localized nuclear pattern co-localization. While this data can indicate that the two proteins participate in common processes, it is not clear to me if it is proof enough to support the idea that *Ythdc2* recruits *Kdm4a* to particular genomic loci.

Thanks to point out the weakness of our evidences. We now have added two sets of new experiments (ChIP & PCA) to further support our conclusion. To further confirm that *Ythdc2* function as a recruiter to assign *Kdm4a* to specific genome sequence, we

performed chromatin immunoprecipitation (ChIP) RT-qPCR analysis and revealed that the Flag-Kdm4a binding of exonic regions of gene *Trpm7* in the Ythdc2 KD Neuro-2a cells was notably lower than that in the control cells (**Response to Reviewer Figure 13a**). Conversely, there were no differences between Ythdc2 KD and control cells in the Flag-Kdm4a binding of intronic regions of gene *Trpm7*. This data suggested that loss of Ythdc2 decreases the deposition of Kdm4a to the exonic region of its downstream genes.

Additionally, we developed a split luciferase-based protein-fragment complementation assay (PCA) to study the Kdm4a-Ythdc2 interactions *in vivo* (**Response to Reviewer Figure 13b**). The full-length firefly luciferase can split into two domains: N-terminal (Nluc, amino acid 2-416) and C-terminal (Cluc, amino acid 389-550). These nonfunctional fragments can fold into an active enzyme when brought into close proximity by the interacting proteins which they were fused. The split luciferase fragments Nluc and Cluc were fused to Kdm4a and Ythdc2 respectively (**Response to Reviewer Figure 13c**). As expected, the fusions expressed alone have no luminescent activity when compared to the untransfected cells (**Response to Reviewer Figure 13d**). Moreover, neither of the noninteracting protein pairs mCherry-Nluc with Ythdc2-Cluc or Nluc-mCherry with Cluc-Ythdc2, resulted in detectable luminescence indicating that the split-luciferase fragments do not spontaneously assembled. Coexpression of Kdm4a-Nluc with Cluc-Ythdc2 or Ythdc2-Cluc resulted in a weak signal above the background, indicating that the positions of luciferase fragments may affect the luminescence intensity which has been previously reported¹⁵. The double PHD and tandem Tudor domains of Kdm4a mediate protein-protein interactions (PPIs), and present in the C-terminal region of Kdm4a^{16,17}. The fusions of Nluc fragment to the C-terminal of Kdm4a could result in the blocking the PPI domains, which might explain why the weak luminescence was detected. Remarkably, a significant luminescence signal was detected when Nluc-Kdm4a was coexpressed with Ythdc2-Cluc or Cluc-Ythdc2, confirming that Kdm4a readily interacts with Ythdc2 *in vivo*. Together, these data confirm that Ythdc2 recruit Kdm4a to form an interacting complex for subsequent transcriptional regulation.

Response to Reviewer Figure 13. (also see Fig. 3I-o)

a, ChIP-qPCR assay for Flag-Kdm4a at the exonic and intronic regions of gene *Trpm7* in WT or Ythdc2 KD cells. **b**, Schematics of the split luciferase complementation assay. **c**, Schematic illustrations of the fused protein variants. **d**, Relative luminescence intensity of each pair of interacting partners.

7- Along the manuscript, molecular biology experiments are conducted indistinctly with different cell lines, including HEK293T, U2-OS and Neuro2A. Is there a particular reason for which that is performed in this way?

In our study, the 293T cells were used for identification of KDM4A-associated proteome in the nucleus. The 293T cell line is suitable for transfection of plasmids and stable expression of fusion proteins. The U2-OS cells were used for visualization of the colocalization of Flag-tagged KDM4A and HA-tagged YTHDC2 proteins in the nucleus. The U2-OS cell morphology is flattened and their nuclei are significantly larger than those of other cell types, such as HeLa or 293T cells, suggesting U-2 OS cell line is more suitable for the nucleus imaging. The Neuro-2a cells are mouse neuroblast-derived cell line. In our study, the Neuro-2a cell line was used for studying regulation of mouse gene expression, such as gene perturbation (validation of the knockout/knockdown efficiency of sgRNAs or shRNAs targeting mouse gene), and RNA transcription (in vitro nuclear run-on assay, dual luciferase reporter assay and split luciferase complementation assay).

8- At statistical level, I have concerns with the number of replicates used for some analyses. For instance, some experiments are presented with N=2 and unpaired two-tailed t-test statistics are still applied (Figure 3i, Extended Data Fig. 3c) while in other cases the number of replicates doesn't seem to match with the number of samples shown in the graph, making difficult to know if the statistics are based on replicates or slices/samples (Extended Figure 2f for instance). Could the authors revise this?

We thank the reviewer for your concerns and suggestions regarding the number of replicates and statistical analysis in our previous manuscript. Sorry for the confusion. We now have revised our data and updated the corresponding figure legends in the latest version of manuscript.

Minor points:

1- In the results section, line 82, the authors comment that the dosage of lentivirus was adjusted to ensure that each infected neuron express a single sgRNA from the library, but no information of how this is achieved or validated is provided to support this. Could the authors elaborate a bit more regarding this?

The dosage was adjusted according to previous sgRNA CRISPR-based screening reports^{18,19} to express red fluorescent protein and sgRNA from the library for enough expression and coverage *in vivo*.

2- I have found several typos along the manuscript that I think would require attention, including some phrases that are difficult to interpret or not adequately structured for a scientific article. I add some examples for the authors to check:

Line 30: "in many brain regions, that distinct cell..." -> "in many brain regions, where distinct cell..."

Line 47: “neuronal activities” -> “neuronal activity”

Line 58: “neuronal activities and track engram activities” -> “neuronal activity and track engram activity”

Line 103: “we designed shRNA targeting on Kdm4a” -> “we designed shRNA targeting Kdm4a”

Line 153: “changes via through RNA” -> “changes through RNA”

Line 179-182: Revise text

Line 228: “Fig. 3kand” -> “Fig. 3k and”

Line 234: “In the same time” -> “At the same time”

Line 285: “Therefore, Kdm4a-removel-mediated” -> “Therefore, Kdm4a-removal-mediated”

Line 334: “a priming state that neurons” -> “a priming state in which neurons”

Line 360: “It was known” -> “It is known”

We appreciate for your helpful corrections and patience to our textual errors. The sentences and wording mentioned above have been revised. The manuscript was rewritten and modified accordingly.

References

1. Cai, D. J. *et al.* A shared neural ensemble links distinct contextual memories encoded close in time. *Nature* **534**, 115–118 (2016).
2. Han, J. H. *et al.* Neuronal competition and selection during memory formation. *Science (80-.)*. **316**, 457–460 (2007).
3. Cai, D. J. *et al.* A shared neural ensemble links distinct contextual memories encoded close in time. *Nature* **534**, 115–118 (2016).
4. Shen, Y. *et al.* CCR5 closes the temporal window for memory linking. *Nature* **606**, 146–152 (2022).
5. Xie, H. *et al.* In vivo imaging of immediate early gene expression reveals layer-specific memory traces in the mammalian brain. *Proc. Natl. Acad. Sci. U. S. A.* **111**, 2788–2793 (2014).
6. Wang, G. *et al.* Switching From Fear to No Fear by Different Neural Ensembles in Mouse Retrosplenial Cortex. *Cereb. Cortex* **29**, 5085–5097 (2019).
7. Wang, G. *et al.* Egr1-EGFP transgenic mouse allows in vivo recording of Egr1 expression and neural activity. *J. Neurosci. Methods* **363**, 109350 (2021).
8. Ding, X. *et al.* Activity-induced histone modifications govern Neurexin-1 mRNA splicing and memory preservation. *Nat. Neurosci.* **20**, 690–699 (2017).
9. Chen, Q., Ding, X., Guo, X., Zhou, G. & Guan, J. Suv39h1 regulates memory stability by inhibiting the expression of *Shank1* in hippocampal newborn neurons. *Eur. J. Neurosci.* **55**, 1424–1441 (2022).
10. Picelli, S. *et al.* Full-length RNA-seq from single cells using Smart-seq2. *Nat. Protoc.* **9**, 171–181 (2014).
11. Yan, Y. *et al.* ASH1L haploinsufficiency results in autistic-like phenotypes in mice and links Eph receptor gene to autism spectrum disorder. *Neuron* **110**, 1156-1172.e9 (2022).
12. Liu, Y. *et al.* TRPM7 Is Required for Normal Synapse Density, Learning, and Memory at Different Developmental Stages. *Cell Rep.* **23**, 3480–3491 (2018).
13. Jiang, Z.-J. *et al.* TRPM7 is critical for short-term synaptic depression by regulating synaptic vesicle endocytosis. *Elife* **10**, (2021).
14. Zhang, Y. *et al.* An RNA-Sequencing Transcriptome and Splicing Database of Glia, Neurons, and Vascular Cells of the Cerebral Cortex. *J. Neurosci.* **34**, 11929–11947 (2014).
15. Li, Z. *et al.* Existing drugs as broad-spectrum and potent inhibitors for Zika virus by targeting NS2B-NS3 interaction. *Cell Res.* **27**, 1046–1064 (2017).
16. Lee, J., Thompson, J. R., Botuyan, M. V. & Mer, G. Distinct binding modes specify the recognition of methylated histones H3K4 and H4K20 by JMJD2A-tudor. *Nat. Struct. Mol. Biol.* **15**, 109–111 (2008).
17. Su, Z. *et al.* Reader domain specificity and lysine demethylase-4 family function. *Nat. Commun.* **7**, (2016).
18. Liu, J. *et al.* A genome-scale CRISPR-Cas9 screening in myeloma cells identifies regulators of immunomodulatory drug sensitivity. *Leukemia* **33**, 171–

- 180 (2019).
19. Wertz, M. H. *et al.* Genome-wide In Vivo CNS Screening Identifies Genes that Modify CNS Neuronal Survival and mHTT Toxicity. *Neuron* **106**, 76-89.e8 (2020).

REVIEWERS' COMMENTS

Reviewer #1 (Remarks to the Author):

The authors have answered my questions/comments to a good level, thereby clarifying this complex study. I also note the inclusion of an additional experiment that aims to investigate the role of Kdm4a in memory allocation. This reveals an intriguing complementary (or related) pathway to one mediated by CREB (and this is how the authors should conclude on the result/discussion). In addition, I have a few of minor comments:

- Line 395 change theregy to thereby
- Rewrite lines 337-350 as the text is confusing especially the following: ..'resulting a pool discrimination...'; 'To further confirmed the memory allocation phenomeone in the memory of'. The text in response to Referees letter is much clearer.

Reviewer #2 (Remarks to the Author):

The revision carried out by the authors has addressed most of the concerns I raised during my initial evaluation of the manuscript and has consistently improved the overall quality of the work, resulting in a more robust study than the initially submitted one. The authors have provided a considerable amount of experimental data to answer my initial concerns and have improved the explanations for some of the results and methods. I think in the present format the study represents an interesting piece of work that will be of great interest for the broad neuroscience community.

I have just observed some typos in the new incorporated text that should be revised by the authors for the final article formatting, as for instance:

Line 99: (< 1%) and glia cell population (< 3%) were also labeled (Extended Data Fig. 3). But they might not have strong impact on this screening due to the limited number. ◊ The two phrases could be combined

Line 126: We found that reducing Kdm4a expression did not significantly increase in the total ◊ We found that reducing Kdm4a expression did not significantly increase the total

Line: 153: To further confirm the Kdm4a is down-regulated in the fear memory-activated ◊ To further confirm that Kdm4a is down-regulated in the fear memory-activated

Response to the referees' comments

We sincerely thank the reviewers and editor for the constructive suggestions and positive response throughout the review process. We are delighted that we have “answered my questions/comments to a good level” and “consistently improved the overall quality of the work”.

Reviewer #1 (Remarks to the Author):

The authors have answered my questions/comments to a good level, thereby clarifying this complex study. I also note the inclusion of an additional experiment that aims to investigate the role of Kdm4a in memory allocation.

We thank the reviewer for the positive feedback on our revisions.

This reveals an intriguing complementary (or related) pathway to one mediated by CREB (and this is how the authors should conclude on the result/discussion).

We have added this point in our revised discussion.

In addition, I have a few of minor comments:

- Line 395 change theregy to thereby*
- Rewrite lines 337-350 as the text is confusing especially the following: ..'resulting a pool discrimination...' ; 'To further confirmed the memory allocation phenomeone in the memory of'. The text in response to Referees letter is much clearer.*

Thanks for pointing this out, we have now corrected it in our revised text.

Reviewer #2 (Remarks to the Author):

The revision carried out by the authors has addressed most of the concerns I raised during my initial evaluation of the manuscript and has consistently improved the overall quality of the work, resulting in a more robust study than the initially submitted one. The authors have provided a considerable amount of experimental data to answer my initial concerns and have improved the explanations for some of the results and methods. I think in the present format the study represents an interesting piece of work that will be of great interest for the broad neuroscience community.

We appreciate the reviewer for the positive response on our revisions and acknowledging that our study “represents an interesting piece of work that will be of great interest for the broad neuroscience community”.

I have just observed some typos in the new incorporated text that should be revised by the authors for the final article formatting, as for instance:

Line 99: (< 1%) and glia cell population (< 3%) were also labeled (Extended Data Fig. 3). But they might not have strong impact on this screening due to the limited number. ◇ The two phrases could be combined

*Line 126: We found that reducing Kdm4a expression did not significantly increase in the total
◇ We found that reducing Kdm4a expression did not significantly increase the total*

Line: 153: To further confirm the Kdm4a is down-regulated in the fear memory-activated ◇ To further confirm that Kdm4a is down-regulated in the fear memory-activated

Thanks for pointing out these errors, we have corrected it in our revised manuscript.